

**Representing anthropogenic gross land use change, wood harvest and forest**
**age dynamics in a global vegetation model ORCHIDEE-MICT (r4259)**
Chao Yue[1], Philippe Ciais[1], Sebastiaan Luyssaert[2], Wei Li[1], Matthew J. McGrath[1], Jinfeng Chang[3],
Shushi Peng[4]
[1]Laboratoire des Sciences du Climat et de l'Environnement, LSCE/IPSL, CEA-CNRS-UVSQ, Université
Paris-Saclay, F-91191 Gif-sur-Yvette, France
[2]Department of Ecological Sciences, Vrije Universiteit Amsterdam, Amsterdam 1081 HV, The
Netherlands,
[3]Sorbonne Universities (UPMC, Univ Paris 06)-CNRS-IRD-MNHN, LOCEAN/IPSL, 4 place Jussieu,
75005 Paris, France
[4]Department of Ecology, College of Urban and Environmental Sciences, Peking University, Beijing
100871, China
*Correspondence to:* Chao Yue (chao.yue@lsce.ipsl.fr)
**Abstract**
Land use change (LUC) is a fundamental anthropogenic disturbance in the global carbon cycle. Here we
present model developments in a global dynamic vegetation model ORCHIDEE-MICT for more realistic
representation of LUC processes. First, we included gross land use change (primarily shifting cultivation)
and forest wood harvest in addition to net land use change. Second, we included sub-grid even-aged land
cohorts to represent secondary forests, and to keep track of the age of agricultural lands since LUC, which
are associated with variable soil carbon stocks. Combination of these two features allows simulating
shifting cultivation with a short rotation length involving mainly secondary forests instead of primary
ones. This is in contrast with the traditional approach where a single patch is used for a given land cover
type in a model grid cell and forests are thus close to primary ones. We have tested the model over
Southern Africa for the period 1501–2005 forced by a historical land use change data set. Including gross
land use change and wood harvest has increased LUC emissions in both simulations with ($S_{age}$) and
without ($S_{ageless}$) sub-grid secondary forests, but larger increase is found in $S_{ageless}$ (by a factor of 2) than
$S_{age}$ (by a factor of 1.5). Emissions from bi-directional land turnover alone are 35% lower in $S_{age}$ than
$S_{ageless}$, mainly because the secondary forests cleared for agricultural land have a lower aboveground
biomass than primary ones. We argue that, without representing sub-grid land cohort demography, the



additional emissions from land turnover / gross land use change are overestimated. In addition, our
developments provide possibilities to account for continental or global forest demographic change
resulting from past anthropogenic and natural disturbances.
Keywords: dynamic vegetation model, gross land use change, age dynamics, shifting cultivation, land use
emissions

**1 Introduction**
Land use and land use change (LUC) strongly modifies the properties of the Earth's surface, ecosystem
services and their carbon and nutrient fluxes. These activities have significant impacts on the Earth's
climate through both biogeochemical and biophysical effects (Foley et al., 2005; Luyssaert et al., 2014;
Mahmood et al., 2014). When a forest is cleared, the majority of carbon stored in the aboveground
biomass is lost as $CO_2$ to the atmosphere. Such loss can occur within a few years when fire is used in
deforestation (Morton et al., 2008), or more slowly through decomposition of the slash left on the ground
(Houghton, 1999). Harvested woods for long-term use, though, often take a few decades to degrade
(Mason Earles et al., 2012). In addition, LUC changes the balance between litter input and heterotrophic
respiration, resulting in changes in soil organic carbon (SOC). A number of meta-analyses (Don et al.,
2011; Guo and Gifford, 2002; Poeplau et al., 2011; Powers et al., 2011) have examined SOC change
following LUC. Though the directions of SOC change are roughly consistent among typical LUC
transitions (e.g., SOC decreases when a forest is converted to cropland; SOC increases when a cropland is
converted to pasture), large uncertainties remain regarding the magnitude of SOC changes and its
relationship with secondary ecosystem management, climate, soil physical and biogeochemical
properties, and the time elapsed since LUC.

Globally, LUC activities have contributed significantly to historical anthropogenic carbon emissions. It is
estimated that about 8 $Mkm^2$ of forests were cleared for agricultural purpose and that 20 $Mkm^2$ of forests
were harvested during 1850–1999, giving rise to cumulative emissions of 124 Pg C, or 33% of the total
anthropogenic emissions (Houghton, 1999). Houghton et al. (2012) reviewed LUC emissions from
multiple studies and estimated global LUC emissions to be 1.1 Pg C $yr^{-1}$ for 1980–2009, with an
uncertainty of 0.5 Pg C $yr^{-1}$. Different estimations of historical LUC emissions by Dynamic Global
Vegetation Models (DGVM) show a spread as large as 1 Pg C $yr^{-1}$ (see Fig. 1 in Houghton et al. 2012; see
also Hansis et al., 2015 for even larger range among estimations). This is partly due to different forcing
data used and initial carbon stocks simulated (Li et al., submitted), but also because of different





implementations of LUC processes in dynamic global vegetation models (Prestele et al., 2016). Given the
importance of historical LUC emissions and its large uncertainty, a more realistic representation of LUC
processes and land management in DGVMs is desirable. This will help improve the diagnostic of the
current global carbon cycle perturbation and better forecast its future evolution, which is useful for
formulating efficient land-based climate mitigation strategies.

In most global studies, only net transitions are accounted for in the LUC processes simulated by DGVMs.
As such, changes in land use for each model grid cell are diagnosed as the difference in ground fractions
of different land cover types between two consecutive years. At a typical spatial resolution of 0.5° for
global applications (e.g., TRENDY, Sitch et al., 2015; MsTMIP,
http://nacp.ornl.gov/MsTMIP_simulations.shtml), such a scheme ignores the simultaneous transitions of
opposite signs between two vegetation types within the same grid cell (i.e., gross transitions). A typical
example is shifting cultivation, which involves clearing a forest for a non-permanent cropland. After the
cropland is maintained for some time, it is laid fallow to allow forest recovery, and farmers then search
for other forests to reinitiate the cycle. Shifting cultivation was historically important in many tropical
regions for the subsistence of its inhabitants (Hurtt et al., 2006; Lanly, 1985). Forest management such as
clear-cut for wood followed by replanting trees is another type of gross transition. Although it does not
entail a change in land cover (forest remaining forest), species choice and forest management can have a
significant effect on carbon stocks and fluxes.

Recently, gross land cover transitions were included in an emulator of the JSBACH DGVM
(Wilkenskjeld et al., 2014) and in the LPX-Bern 1.0 DGVM (Stocker et al., 2014). Both studies reported
additional LUC carbon emissions when including gross transitions, largely due to the imbalance between
moderate carbon uptake in recovering fallow lands and large carbon release from recently cleared lands.
Despite promising results of these two studies, among the DGVMs used to assess LUC emissions in the
annual update of global carbon budget (Le Quéré et al., 2016), none of them included gross LUC and only
a few included wood harvest.

One must keep in mind, however, that omitting sub-grid gross transitions in DGVMs is largely a scale-
dependent issue — suppose DGVMs could run at any finite spatial resolution, then all transitions would
be net ones. But given the typical coarse spatial resolution at which DGVMs are often applied, specific
routines are needed in the model to include these gross transitions. Another highly relevant aspect is that,
DGVMs often use abstract patches associated with certain fractions of a grid cell to represent different
land cover types. In most cases only a single patch is used for a certain land cover type. As a result, sub-



grid age structure within the same land cover type cannot be represented. When a new forest patch is
created due to agricultural abandonment, or re-grows after a clear-cut event, in a DGVM this young forest
patch has to be first established and then numerically merged with the existing patch of the same forest
type. The carbon stocks are averaged as well, following an area-weighted approach (known as the
"dilution approach"). This has an implication for simulated LUC emissions. For example, if shifting
cultivation has a rotation cycle of several years to a few decades, the carbon density of the cleared forest
will be smaller if a secondary forest is explicitly simulated and cleared, compared to the approach of a
single patch representation, where the forest cleared has no age and possibly has high carbon stocks. This
calls for inclusion of sub-grid cohorts in DGVMs when simulating gross land use change and forest
management. Some recent developments of DGVMs have included this aspect for both forest
management and certain LUC processes (Naudts et al., 2015).

For the reasons described above, the objectives of this manuscript are: (1) to describe the development of
a new LUC module, including sub-grid vegetation cohorts, forest harvest and gross land use change in the
ORCHIDEE DGVM, that can be run with and without sub-grid sub-grid age dynamics; (2) to document
the model behavior, and (3) to test the hypothesis that including gross transitions and harvest increases the
simulated LUC emissions, but that these additional emissions tend to be over-estimated when sub-grid
age dynamics are not accounted for.
**2 Methods**
**2.1 Model developments to include sub-grid vegetation cohorts and gross transitions**
**2.1.1 Original land use change module with net transitions only**
The model version as the starting point for our development is ORCHIDEE-MICT (r3247), a branch of
the ORCHIDEEE DGVM (the major version is called the trunk version), the land surface component of
the French IPSL Earth System Model (ESM). ORCHIDEE can simulate the energy, water and carbon
fluxes between the land surface and the atmosphere. The carbon module simulates vegetation carbon
cycle processes, including photosynthesis, photosynthates allocation, vegetation mortality and
recruitment, phenology, litter fall and soil carbon decomposition. ORCHIDEE-MICT is a branch initially
focusing on improving high-latitude processes (e.g., soil freezing, snow processes, permafrost dynamics
and northern wetland) but is now under development to include more processes. Notably, the grassland
management module developed in Chang et al. (2013) is included (r2615). This allows for distinction
between natural grassland and pasture when simulating historical land use change.


In ORCHIDEE, land cover types are represented as plant function types (PFTs), with each PFT being
associated with a set of parameters. A typical model simulation consists of two stages: a spin-up stage
with stable or constant forcing data, where the model is run until an approximately equilibrium state is
reached, to mimic an era with no appreciable human perturbation, and a transient stage, where the model
is forced with temporally varying forgings (e.g., climate, atmospheric $CO_2$, land cover etc.). The land use
change module prior to this study accounts for net transitions only (Piao et al., 2009a) and has been used
in many applications (e.g., CMIP5, http://icmc.ipsl.fr/index.php/cmip5; TRENDY, Sitch et al., 2015). To
simulate historical land use change, a spin-up stage is started with a given initial land cover map (i.e., a
PFT map), and then vegetation distribution is updated annually with prescribed PFT map time series
during the transient simulation. The LUC module simply compares grid cell fractions of different PFTs
between the current simulation year and the next year. Then twelve vegetative PFTs (all standard model
PFTS excluding the bare soil PFT) are separated into two groups with expanding versus contracting areas.
Carbon stocks and associated carbon fluxes on shrinking PFTs are displaced to expanding PFTs in
proportion to their respective surface increments.
**2.1.2 Concept of gross transitions in relation to vegetation age structure**
The numerical implementation of net transitions is straightforward. However, as explained in the
introduction, this scheme omits important sub-grid gross land use transitions. Figure 1 uses an exemplary
grid cell to illustrate the difference between the two LUC schemes: one accounting for net transitions only
(Fig. 1b), and the other accounting for gross transitions but with no sub-grid cohorts (Fig. 1c & 1d).
Although the areas of forest and cropland after LUC are identical (Fig. 1b & 1d), carbon stocks for the
same vegetation type (e.g., forest) are different between the two schemes. According to the net transition
scheme, the carbon stock of the final forest patch shown in Fig. 1b remains intact. But under the gross
scheme (Fig. 1d), the post-LUC forest carbon stock is an area-weighted mean between the original forest
patch not impacted by LUC, and the newly established forest with a low carbon density that results from
cropland abandonment. Consequently the carbon stock of the grid cell is expected to be smaller in Fig. 1d
than in 1b. LUC carbon emission in Fig 1d is conversely larger than in 1b.

Figure 1c represents the real land cover state after LUC, while the dilution shown in Fig. 1d is only a
necessary simplification when no sub-grid cohorts are represented in the model. Ideally, the model
capacity could be expanded to include cohorts, to represent the real world case as in Fig. 1c. In addition,
inclusion of sub-grid cohorts would allow not only the distinction between original intact forest and
newly established forest, but also allow distinguishing among different forest cohorts (e.g., primary
versus secondary forests) regarding the decision on which forest should be cleared for cropland.





Figure 2 illustrates a case where gross LUC is combined with sub-grid cohort representation in the model,
which allows accommodating land use transitions in a more realistic way. Here, multiple patches within a
grid cell are used to represent cohorts of a single vegetation type but with different ages since
establishment. These cohorts often have different carbon stocks either due to different lengths in carbon
accumulation time (e.g., for forest) or due to different extents to which the legacy soil carbon is reserved
(e.g., for cropland establishing on former forest). The areas subject to gross LUC transition in Fig. 2a &
2b remain the same as in Fig. 1a (dashed red rectangles), but primary and secondary forests are cleared in
Fig. 2a and Fig. 2b, respectively. Thus LUC emissions from clearing of primary forest are expected to be
higher due to its higher biomass stock. Correspondingly, the legacy soil carbon stocks on the cohort of
new cropland are also higher (shown in Fig. 2b & 2d).

Figure 1 and Fig. 2 have shown the example of LUC transitions between forest and cropland, but other
types of land use changes, including forest harvest, can be handled in a similar way. In the case of forest
harvest, such as in Fig. 2, having cohorts avoids the simplification to merge a young re-established forest
after harvest with the original forest, which serves as the exact source of harvest. This can effectively
simulate forest management practices inducing rotations between different forest cohorts that vary with
time (e.g., see McGrath et al., 2015 for forest management history in Europe).

**2.1.3 Expansion of ORCHIDEE-MICT capacity to represent sub-grid vegetation cohorts**


In order to simulate gross LUC combined with sub-grid vegetation cohorts as illustrated in Fig. 2, we
expanded the ORCHIDEE-MICT capability to include sub-grid even-aged cohorts. This necessitates
multiple patches within a grid cell for a single PFT, which inherit most of the parameters from their
parent PFT (they still belong to the same PFT and thus are largely physically similar). These patches are
named here *Cohort Functional Types (CFT),* to be distinguished from the original *plant functional types*.
In this sense, the original PFTs actually become "meta-PFTs" which we label as meta-classes (MTCs). As
subsequent land use changes greatly increase the total number of CFTs, the computational demand will be
greatly increased. Hence, the number of CFTs within an MTC is limited to a user-defined number.

ORCHIDEE-trunk has a feature called "PFT externalization" which allows creating a user-specified new
PFT by inheriting its parameters from an existing one whose parameterization is well defined. A user can
then modify specific parameters at their convenience. Based on this feature, the ORCHIDEE-CAN (svn
rev. = r2566; Naudts et al., 2015, Page 2037) branch incorporated representation of sub-grid forest age
classes (i.e., equivalent to our CFTs here) for European forests. Each forest age class is an inheritance of a
given forest MTC. There, forest age classes were defined by different tree diameters. When a forest of a
certain age class reaches its diameter limit, it moves into the next age class, and is merged with the





existing forest patch if there is one. All associated biophysical and biogeochemical variables are merged
as well following an area-weighted mean approach.

ORCHIDEE-MICT also inherits this "externalization" feature from ORCHIDEE-trunk. Here we ported
the codes of forest age class functionality from ORCHIDEE-CAN, and made all necessary adaptions into
ORCHIDEE-MICT, to develop the CFT functionality needed for LUC simulation with cohorts. Forest
canopy structure and tree diameters are simulated in ORCHIDEE-CAN, using an allometry-based
allocation scheme (based on the pipe model) and a tree-height dependent light attenuation scheme
(Naudts et al., 2015, Page 2038). ORCHIDEE-MICT, however, uses the same big-leaf approximation and
exponential attenuation of light in the canopy as in ORCHIDEE-trunk to scale photosynthesis from leaf to
canopy depth (Krinner et al., 2005). As a result, no tree diameter classes exist in ORCHIDEE-MICT and
we thus use forest woody biomass to delimit different cohorts, with older cohorts having a higher woody
biomass. In addition, we expanded the concept of CFT to croplands, natural grasslands and pastures.
Cohorts are defined with their soil carbon stocks for these herbaceous vegetation types; this is a definition
relevant to LUC emission calculation. For these short-vegetation CFTs, we assume that the older their age
since LUC disturbance, the lower their soil carbon will be (assuming a typical case of cropland
originating from forest). The biomass or soil carbon thresholds that delineate different CFTs must be
properly parameterized in order to have sensible CFT segregation within different contexts of land use
change. This will be further detailed in the Sect 2.2.3. In practice, for single-site simulation, the
parameterization could be set up via a configuration file enumerating the thresholds for all CFTs. For
regional applications, an input file containing thresholds for each grid cell will be used.

The implementation of sub-grid cohort function types as inheritances of meta-classes and the
corresponding hierarchy are exhibited in Fig. 3a. "Tier 1" of the "*Model parameterization hierarchy*"
corresponds to the four basic vegetation types (forest, natural grassland, pasture, and croplands,
abbreviated as *f, g, p, c* respectively). "Tier 2" corresponds to meta-classes in ORCHIDEE-MICT, which
contain one bare soil MTC and fourteen vegetative MTCs, with each vegetative MTC belonging to one of
the four basic vegetation types. "Tier 3" corresponds to cohort function types. A cohort functional type is
conveniently noted as $CFT_{i,j}$ to denote that it inherits its parameter values from the $MTC_i$ and belongs to
the $j^{th}$ cohort. Forest MTCs contain six CFTs and herbaceous MTCs contain two CFTs. The number of
CFTs for each MTC is not hard-coded in the model and can be specified by users via a configuration file.

With sub-grid cohorts, the model spin-up run is initiated with an input MTC map, essentially the same as
in the case without sub-grid cohorts (recall that in Sect. 2.1.1 this MTC map is called a PFT map). But the





difference is that the initial prescribed areas (as fractions of grid cell area) of different MTCs are assigned
to their youngest cohorts. With model spin-up going on, forest woody mass will grow to exceed the
thresholds of the first cohort, so that forests will move to the second cohort, and so on. At the end of spin-
up, all forests thus end up in the oldest cohort of each MTC. The same case applies to herbaceous MTCs,
given that cohort thresholds are properly defined.

Natural forest mortality in ORCHIDEE could be either prescribed as a constant rate or dynamically
simulated, but the mortality process only has an effect to reduce the amount of existing biomass in each
forest cohort, when dead biomass is moved to litter pool and recruitment carbon stocks are integrated.
This remains the same as the case without sub-grid cohorts, i.e., natural recruitments do not create young
cohorts but just "dilute" the carbon stock of each forest cohort. Open vegetation fires are handled in a
similar manner. ORCHIDEE-MICT has integrated a prognostic fire module (Yue et al., 2014) to simulate
open grassland and forest fires arising from both natural and anthropogenic ignitions. Fire-induced forest
mortality is handled similarly as natural mortality, i.e., fire-induced recruitments lead to no young cohort
creation but just reduce the existing carbon stock. Other forest disturbances, such as wind-throw, diseases
and insect outbreaks, are not explicitly considered. Because of these reasons, after the spin-up, the only
way to create secondary cohorts is through land use change.

When entering transient simulations with land use change being included, younger cohorts will begin to
be created as a result of different LUC activities. From a modeling perspective, the oldest cohorts in
ORCHIDEE-MICT are somewhat equivalent to the primary lands (especially, the oldest forest cohorts
are equivalent to primary forests), and other younger cohorts are analogue to secondary lands.
**2.1.4 Model developments to include gross land use change and forest harvest, with and without**
**sub-grid cohorts**
This section describes the implementation of gross land use change and forest harvest with sub-grid
CFTs. We focus on the implementation with sub-grid cohorts, because the same LUC process without
cohorts could be simply treated as a particular case where all MTCs have only one single cohort. The
module interface is designed to receive forcing information on land area fluxes among four basic land
cover types of forest ($f$), natural grassland ($g$), pasture ($p$) and cropland ($c$), taking into account the current
LUC modeling landscape in DGVMs (as briefly reviewed in the Introduction) and the availability of land-
use change history reconstructions (e.g., Hurtt et al., 2011). In order to compare the simulation results
from the gross LUC module with the original net-transition-only LUC module, we separate the gross
LUC areas into two additive terms: 'net change' equivalent to the original net transition (prescribed by the
matrix $M_{net}$), and 'land turnover' for the bi-directional equal land fluxes between any pair of land cover





types (prescribed by the matrix $M_{turnover}$). Similarly, the forest harvest information is prescribed in a third
matrix $M_{harvest}$. For the moment, information for all the three LUC types is provided as fraction of grid cell
area. This is a deliberate choice, mainly for the convenience of progressive stage-wise model
development. We will come back to the influence of this choice within the land use decision contexts in
the Discussion section.

The key processes of the gross LUC module with CFTs are shown in Fig. 4, comprising in total 6 steps.
The LUC module is called at the first day of each year. Input data are the three matrices. $M_{net}$ and $M_{turnover}$
are both square matrices with a size of 4 by 4:

$$\mathbf{M_{net}\,(M_{turnover})} \;=\; \begin{array}{c} \\ \\ \text{forest} \\ \text{grassland} \\ \text{pasture} \\ \text{cropland} \end{array} \begin{bmatrix} F_{f \triangleright f} & F_{f \triangleright g} & F_{f \triangleright p} & F_{f \triangleright c} \\ F_{g \triangleright f} & F_{g \triangleright g} & F_{f \triangleright p} & F_{g \triangleright c} \\ F_{p \triangleright f} & F_{p \triangleright g} & F_{p \triangleright p} & F_{p \triangleright c} \\ F_{c \triangleright f} & F_{c \triangleright g} & F_{c \triangleright p} & F_{c \triangleright c} \end{bmatrix}$$

Receiving land type
forest grassland pasture cropland
Donating land type

278                                                                                      Eq (1)

Where the element $F_{i>j}$ denotes the land flux from land cover type $i$ to $j$, with $i, j$ being elements of the
vector of $[f\,g\,p\,c]^{\mathbf{T}}$. The diagonal elements correspond to land area intact from any land use transitions
and are simply ignored in the LUC module. By definition, $M_{turnover}$ is a symmetric square matrix. $M_{harvest}$ is
a matrix with only two elements: harvest area from primary and secondary harvest.

As explained in Sect. 2.1.3, the construction of CFTs within the model follows the "model
parameterization hierarchy" shown in Fig. 3a. The cohort age subjected to LUC of is one of the most
important considerations in land use change decisions, especially in the context of land turnover and
forest harvest. This necessitates a re-organization of the CFTs to derive the "*LUC hierarchy*" shown in
Fig. 3b, where Tier 2 information is about areas of different cohorts of the same land cover type, and Tier
3 remains on the level of CFTs. So the Step 1 in the LUC module (Fig. 4) is to construct the "LUC
hierarchy", i.e., to calculate within the model the areas of each cohort for each vegetation type.

When implementing LUC matrices, all information of land transitions between the four basic land cover
types must first be downscaled on the cohort tier (i.e., decision on which cohort is subjected to LUC) and
then on the CFT tier (i.e., how LUC-affected area is distributed among different comprising meta-classes
within each cohort, refer also to Fig. 3b). This is achieved in Step 2 as shown in Fig. 4. Because all the
newly established lands, regardless of their originating LUC process, must belong to the youngest CFT of





the MTCs that comprise the target land cover type, the ultimate outcome of Step 2 is a single (large)
matrix $\mathbf{M_{nCFT,\ nMTC}}$ (nCFT = # of CFTs, nMTC = # of MTCs), which indicates the area transferred from
each CFT to the youngest cohort of the concerning MTC. The rules to convert LUC matrices into
components of $M_{nCFT,\ nMTC}$ depend on LUC types and will be explained in detail later. But as long as Step
2 is done, the remaining steps are rather straightforward.

Step 3 handles forest wood collection (here 'collection' rather than 'harvest' is used, to avoid the
confusion with forest wood harvest which is a means of forest management), from forest being converted
to other land cover types, and forestry harvest (forest remaining forest). We assume that a certain fraction
of aboveground woody biomass (i.e., sapwood and heartwood) is lost as instant $CO_2$ flux into the
atmosphere (i.e., due to on-site disturbance), and that the remaining wood is collected as wood product
pools. Step 4 involves the proper displacement of associated carbon stocks and fluxes from the donating
CFTs to the newly established (youngest) cohorts of MTCs, after wood collection. Notably, the legacy
carbon stocks in litter and soil are collected from the donating CFTs and transferred to the newly
established youngest CFTs. Then in Step 5, each youngest CFT cohort is established and initialized, with
its fraction of grid-cell area being the sum of contributed areas given by each source CFT. Finally, in Step
6, a newly established cohort is merged with the existing youngest CFT cohort if there is already one.
When merging of stocks or fluxes between the newly established and existing CFTs, an area-weighted
mean approach is followed:
$$x_{merged} = \frac{x_{new} \times area_{new} + x_{existing} \times area_{existing}}{area_{new} + area_{existing}} \qquad \text{Eq (2)}$$
Where $x$ is the variable in question (e.g., leaf biomass, soil carbon stock etc.,), $x_{new}$ and $x_{existing}$ are the
values of the newly established patch and existing patch before merging and $x_{merged}$ is the value of the
generated patch after merging. $area_{new}$ and $area_{existing}$ are patch areas of the newly established and
existing patches, respectively.

We now return to Step 2, explaining the different rules used to build the $\mathbf{M_{nCFT,\ nMTC}}$ components for
different LUC types. We start with $\mathbf{M_{harvest}}$ by assuming that it precedes conversion of forest to other land
cover types (i.e., land turnover or net land use change). As is explained, the LUC module is designed to
receive externally prescribed harvest information, especially from the widely used LUH1 reconstruction
(Hurtt et al., 2011), rather than to determine harvest volume internally within in the model, which is very
different from ORCHIDEE-CAN. The LUH data makes distinction between harvests from primary and
secondary forests. The harvest information is provided as both forest area and wood volume. Here we
used the area information (again a deliberate choice which will be discussed in Sect. 4). Because of this,



ensuring the consistency between the harvest area in the forcing and that being actually realized in the
model is an important consideration. Moreover, as we want to compare simulated LUC impacts between
the two model configurations with and without sub-grid cohorts, it is necessary to ensure that exactly the
same LUC area is realized in both configurations. This involves a set of decision rules to properly allocate
the prescribed harvest area into different forest cohorts.

Implementation of primary forest harvest is straightforward: we always start with the oldest cohort and
move sequentially downwards to younger ones if older cohorts are exhausted, until the prescribed harvest
demand is fulfilled. For secondary forest harvest, we start with intermediate-aged cohorts. But if the
existing area of intermediate-aged cohorts is not sufficient to fulfill the prescribed harvest area, we are left
with two options to either search upwards for older cohorts or downwards for younger ones. We decide to
first go first for upward searching and then for downward searching, if all cohorts older than intermediate
age still cannot fulfill the prescribed harvest demand. This rule allows potential temporal changes in
harvested area to be accommodated, as explained in Fig. 5. Under such a scheme, (1) at the very
beginning (after spin-up) and before the existence of any secondary forests, harvest will start with the
oldest cohort, i.e., corresponding to harvest of primary forests (sometimes, because of the inconsistency
between the input harvest information and existing forest cohort structure in the model, "secondary"
forest harvest could be prescribed for pixels where only primary forests exist in the model). (2) If harvest
area of secondary forest remains stable, then as soon as sufficient intermediate-aged cohorts are created
via conversion of primary forest to re-growing younger cohorts, a corresponding stable cycle would be
maintained in the model as well. (3) If the harvest area increases, the upward searching would allow
additional harvest of primary forests (i.e., area subject to the stable rotation cycle is expanded). (4) If the
harvest area decreases, the moving of cohorts from younger to older ones independent of any LUC
activities would allow restoring older cohorts— e.g. a consequence of abandonment of forest
management. (5) Finally, the downward searching for younger cohorts after exhausting all other older
cohorts is solely to ensure the consistency between prescribed input harvest area and that actually realized
in the model. Hence, this scheme is designed in order to faithfully implement the prescribed harvest areas
in the model explicitly considering the forest successional states (i.e., primary or secondary). But when
this is not possible because of inevitable disagreement between the model and forcing data, harvest areas
of primary and secondary forests could mutually compensate for each other in the model, to ensure the
their prescribed total harvest area is till realized.

A number of studies reported that fallow lengths for shifting cultivation could range from a few years to
more than 50 years depending on different regions, with the majority being 10–40 years (Bruun et al.,





2006; Mertz et al., 2008; Thrupp et al., 1997; van Vliet et al., 2012), and there is tendency in reduction of
fallow lengths possibly because of increased population pressure (van Vliet et al., 2012). Hurtt et al.
(2006) assumed a mean residence time of 15 years for shifting cultivation for tropical regions in the
LUH1 reconstruction data. Based on these evidences, we assume forest clearance for shifting cultivation
to occur primarily in secondary forests, and treat it similarly as secondary forest harvest when allocating
the prescribed LUC area into different cohorts. The only difference is that the destination land cover
remains forest in the case of forest harvest but is agricultural land in the case of shifting cultivation. For
all other land transfers in shifting cultivation (e.g., pasture to forest), we start exclusively from the oldest
cohort and move downwards to younger ones. For net land use change, priority is again given to older
cohorts followed by younger ones.

Finally, we still need to downscale the LUC area in each cohort to its component CFTs. This is done by
allocating the LUC area in each cohort to its member CFTs in proportion to the existing area of each CFT.

### 2.1.5 LUC processes that remain unchanged in the model

ORCHIDEE simulates two wood product pools with turnover times of 10 years and 100 years,
respectively. Fractions of aboveground woody biomass as instant on-site loss ($F_{instant}$), and entering into
the two wood product pools ($F_{10yr}$, $F_{100yr}$) follow the values in the original net-transition-only LUC scheme
(Piao et al., 2009a), as shown in Table 1. Other biomass compartments (i.e., leaves, fine roots, coarse
roots, fruits and reserve pool) are transferred to litter pools during forest harvest or deforestation.

Other processes relevant to LUC are left unchanged from the original model version. In particular, crop
harvest is applied to cropland CFTs with 45% of biomass turnover being 'harvested' in the model and
exported outside the ecosystem (Piao et al., 2009a). Pasture CFTs are also harvested in the same fashion.
Fires are simulated with a prognostic module, but as explained in Sect. 2.1.3, fire disturbances do not lead
to creation of young cohorts, but only their carbon consequences (e.g., emissions, vegetation mortality,
etc.) are included.

### 2.2 Simulation set-up

### 2.2.1 Definition of land-use change emissions ($E_{LUC}$) and carbon flux sign convention

The land carbon balance simulated by ORCHIDEE r4259 (i.e., net biome production or NBP), when land
use change is included, is defined as:

$$NBP = NPP + F_{Inst} + F_{Wood} + F_{HR} + F_{Fire} + F_{AH} + F_{Pasture} \qquad Eq (3)$$



Where NPP is the net primary production, and all fluxes with "F" notation are outward carbon fluxes
from the land system (they are assigned a negative sign following the ecosystem convention, indicating
that carbon is lost from ecosystems), with $F_{Inst}$ for the instantaneous carbon flux during LUC (e.g., carbon
release arising from site preparation, land-clearing burning etc.), $F_{Wood}$ for the delayed carbon release due
to wood products degradation, $F_{HR}$ for heterotrophic respiration from litter and soil organic carbon, and
$F_{AH}$ for agricultural harvest on both croplands and pastures (assumed to be released to the atmosphere
within one year), and $F_{Pasture}$ for carbon sources from pastures than harvest, i.e., export of animal
production and methane emissions (see Chang et al., 2015 for details).

The LUC emissions ($E_{LUC}$) are quantified as the difference in simulated NBP between two paired
simulations, with LUC (or a specific LUC process) included in one simulation but not the other one:

$E_{LUC} = NBP_{LUC} - NBP_{control}$               Eq (4)

Where, $NBP_{LUC}$ and $NPB_{control}$ are NBP simulated with and without LUC. A negative $E_{LUC}$ denotes a
carbon source to the atmosphere, i.e., ecosystem carbon sink is reduced because of land use change. This
definition follows Pongratz et al. (2014, Page 178) and is also the same as used in TRENDY (Sitch et al.,
2015) simulations and Le Quéré et al. (2016). As explained by Pongratz et al. (2014), such a definition
quantifies the "net" LUC flux because it integrates both emissions to the atmosphere (e.g., deforestation)
and uptakes by potentially recovering vegetation (e.g., agricultural abandonment). More specifically, this
corresponds to the definition "D3" using uncoupled DGVM simulations in Pongratz et al. (2014, Eq. 15c,
Page 187), which contains instantaneous fluxes, legacy fluxes, and "loss of additional sink (source)
capacity (LOAS)".

Instantaneous fluxes refer to the carbon emissions directly arising from LUC, often occurring within the
first year since LUC ($F_{Inst}$ in our case). Legacy fluxes arise from the readjustment of carbon stocks to the
new type of vegetation and/or type and intensity of management over time (Pongratz et al., 2014), and
"loss of additional sink (source) capacity (LOAS)" refers to the carbon sink/source difference between the
actual land cover after LUC and the otherwise potential one under environmental perturbations. All other
flux terms on the right side of Eq. (3) except $F_{Inst}$ contribute to the legacy fluxes and LOAS. Here, as our
model development mainly distinguishes the biomass carbon of secondary forests and it's thus expected
that $F_{Inst}$ and $F_{Wood}$ will be the major fluxes to have influence on simulated $E_{LUC}$. To facilitate the
demonstration of model behaviour, we refer to $F_{Inst}$ and $F_{Wood}$ collectively to as "LUC-associated direct
fluxes" and their variations will be examined in detail on using an idealized grid cell simulation.




The model developments presented here enable us to make two parallel simulations that include LUC:
with and without sub-grid age dynamics. Their simulated $E_{LUC}$ can thus be compared, to separate the
effect of including sub-grid age dynamics. Henceforth for briefness, we denote the simulation without
sub-grid age dynamics as $S_{ageless}$, and the one with age dynamics as $S_{age}$.

**2.2.2 Idealized simulation on a single grid cell**

We conducted an idealized grid cell simulation with prescribed land cover and LUC matrices, to compare
in detail the simulated carbon pools and fluxes between $S_{age}$ and $S_{ageless}$. The geographical coordinates of
the simulation site are 9.25°S, 18.25°E at a 0.5° global grid, in the north of Angola, Africa, where the
miombo woodlands are known to be subject to practices of shifting cultivation. The ESA CCI land cover
map for the 5-year period of 2003–2007 (https://www.esa-landcover-cci.org/) shows a dominant fraction
of tropical deciduous broadleaf forest for this grid cell. Hence for this idealized experiment, the initial
vegetation composition is prescribed as 85% of tropical deciduous broadleaf forest and 15% of C4
cropland. As we will focus on the LUC impacts, other model forcings (climate, atmospheric $CO_2$, etc.) are
held as constant, with climate input data recycling the year of 1901 (CRUNCEP-v5.3.2 climate data,
https://esgf.extra.cea.fr/thredds/fileServer/store/p529viov/cruncep/readme.html) and atmospheric $CO_2$
concentration being fixed at 350 ppm. The model is tested for a hypothesize scenario of constant annual
land turnover with 5% of grid cell area between forest and C4 cropland. Forest harvest of the same
intensity is expected to have largely similar impact. The spin-up was run for 450 years until biomass and
soil C stocks reached equilibrium and the mean annual net biome production (NBP) was close to zero.
Starting from the spin-up, a transient simulation with the prescribed LUC matrix was performed for 100
years.

**2.2.3 Simulation over Southern Africa**

Subsequently, the model behavior has been documented for a real-world case over the region of Southern
Africa (south to the equator of the African continent). All three LUC types occurred historically in this
region, making it ideal to demonstrate model behavior especially regarding forest cohort dynamics as
presented in Fig. 5. This regional simulation serves a single purpose — to further exemplify model
features that cannot be sufficiently demonstrated on one grid cell.

The regional simulation is done at 2° resolution for 1501–2005. We used land use reconstruction from
LUH1 covering 1501–2013 (Hurtt et al., 2011, http://luh.umd.edu/data.shtml#LUH1_Data) re-gridded
from the original 0.5° to a 2° spatial resolution. We derived from the LUH1 dataset the matrices of the
three types of land use change: net land use change, land turnover and wood harvest. Land turnover





information is extracted from LUH1 as the minimum land fluxes between two vegetation types. Because
all LUC activities are represented with matrices, strict area conservation is ensured when re-gridding a
matrix from a higher to lower spatial resolution. Climate forcing data are from CRUNCEP-v5.3.2 at a 2°
resolution. For the spin-up, climate data were cycled from 1901 to 1910, with atmospheric $CO_2$
concentration fixed at 1750 level (277 ppm). In the transient simulation, atmospheric $CO_2$ concentration
began to increase in 1750, climate data were varied starting 1901. The dynamic vegetation module was
turned off, in order to apply the prescribed historical land use change. Factorial simulations are conducted
to quantify $E_{LUC}$ from each of the three LUC types, as shown in Table 2.

Each forest MTC has six CFTs to represent six cohorts. The woody mass thresholds are set in a way that
they correspond roughly to the woody masses at ages of 3, 9, 15, 30, 50 years, and the mature or primary
forest during the spin-up simulation, respectively, for $Cohort_1$ to $Cohort_6$. The $Cohort_3$ with an age of 15
years is the primary target for secondary forest harvest and land turnover (or shifting cultivation),
corresponding to the mean residence time of 15 years of shifting cultivation assumed in LUH1 (Hurtt et
al., 2006) data. We set two CFTs for each herbaceous MTC with a high and low soil carbon density,
respectively. The CFT thresholds of soil carbon stock are the same for all herbaceous MTCs. We first
calculate the maximum soil carbon stock of all MTCs (including the forest ones) at the end of spin-up for
each grid cell, and cohort thresholds are then taken as this maximum value and its 65% value. Because the
energy balance in ORCHIDEE-MICT is resolved for the average of all CFTs over a grid cell, and the
hydrological balance is resolved for three sub-grid water columns (i.e. the water column of bare soil,
forest and herbaceous vegetation), we expect the factors influencing soil carbon decomposition (e.g., soil
temperature, soil moisture) to have little variation among CFTs of the same MTC. This justifies the small
number of herbaceous CFTs, for the sake of computation efficiency.

## 3 Results

### 3.1 Grid cell simulations with and without sub-grid forest age dynamics

#### 3.1.1 Temporal patterns of biomass carbon stock during the spin-up and transient simulations

Figure 6a and 6b exhibit the evolution of above- and belowground biomass for both $S_{ageless}$ and $S_{age}$
simulations, for the spin-up and transient simulation for a test grid cell located in Angola. For this test an
annual forest-cropland turnover of 5% of the grid cell area was imposed. Figure 6c shows changes in the
ground fractional cover of different forest cohorts in $S_{age}$ during the transient simulation. $S_{ageless}$ and $S_{age}$
share the same biomass accretion with time during the spin-up, but $S_{age}$ shows a succession of forest
cohorts — with biomass moving from one cohort to the next (Fig. 6a & 6b). At the end of the spin-up, all
biomass is found in $Cohort_6$ (i.e., the oldest cohort) in $S_{age}$, with an initial forest cover of 85%.






More differences emerge when entering the transient simulation. Aboveground biomass in $S_{ageless}$ shows
an initial sharp drop followed by a more gradual decline under constant land turnover, because biomass of
the single forest patch is constantly 'diluted' by merging with the new forest patch with a low biomass,
which is established out of land turnover (see also Fig. 1). Belowground biomass, however, shows a
corresponding initial drop but then slightly increases. Eventually, both above- and belowground biomass
stocks in $S_{ageless}$ reach a new equilibrium, which are lower than their values at the end of the spin-up. By
contrast, in $S_{age}$, the fraction of $Cohort_6$ declines with the start of the transient simulation because of
conversion to cropland. This decline continues until the $12^{th}$ year, after which the remaining $Cohort_6$
covers only 30% of the grid cell (Fig. 6h). Younger cohorts are progressively created as forests restore
after shifting agriculture abandonment, with the $Cohort_1$ (i.e., the youngest one) appearing during the
initial 6 years after the start of LUC, after which its biomass is moved into $Cohort_2$ (Fig. 6c & 6d).
$Cohort_3$ starts to appear at the $12^{th}$ year when biomass in $Cohort_2$ moves into it. Then its coverage declines
as this cohort, rather than $Cohort_6$, is used as the source for shifting cropland, according to the model rule
that secondary forest is taken prior to primary forest in the land turnover (Fig. 5). After the initial 15 years
(the rough age of $Cohort_3$), the fractions of $Cohort_1$, $Cohort_2$ and $Cohort_3$ reach a dynamic stable state.

While the aboveground biomass continuously grows during the spin-up, the belowground biomass first
increases with time and then slightly declines before reaching the equilibrium value. This is because
ORCHIDEE-MICT has a preferential allocation of NPP allocation to belowground sapwood when forests
are young. The small decline in belowground biomass in the late spin-up stage thus results from an almost
stabilized NPP (under a big-leaf approximation), the reduced belowground allocation and a constant
mortality. Because of this feature, ORCHIDEE-MICT creates a higher belowground biomass in younger
forest cohorts (e.g., $Cohort_2$ and $Cohort_3$ in Fig. 6a & 6b) in $S_{age}$ than the single forest patch in $S_{ageless}$ in
the transient simulation. However, the aboveground biomass in younger $Cohort_2$ and $Cohort_3$ in $S_{age}$ is
lower than $S_{ageless}$. The difference in biomass influences the simulated $E_{LUC}$ between these two
simulations, as we will discuss in detail later.
**3.1.2 LUC-associated direct carbon fluxes**
As shown in Fig. 7a, in $S_{ageless}$, the instantaneous carbon flux resulting from LUC follows the same
temporal pattern than the aboveground biomass, as it is simulated as a fixed fraction of aboveground
woody mass (sapwood and heartwood) (see Sect. 2.1.5). In $S_{age}$, for the initial 12 years, the $Cohort_6$
(undisturbed mature forest) is cleared, so that the instantaneous LUC carbon flux is higher than that in
$S_{ageless}$ (where the biomass of the single forest patch is "diluted" immediately after the land turnover
starts). After that, the instantaneous flux shows a stark drop in $S_{age}$ when the $Cohort_3$ enters the land



turnover. Since then until the end of the simulation, $S_{age}$ has kept a constantly lower instantaneous flux
than $S_{ageless}$ because the LUC-perturbed equilibrium biomass in the latter case is higher (Fig. 6a).  As a
fixed 10% of aboveground woody biomass enters the wood product pool with a 10-year turnover time,
delayed carbon emissions from wood products degradation in both simulations are smaller than the
instantaneous LUC carbon fluxes. They peak around the $12^{th}$ year after LUC and remain stable afterwards
(Fig. 7a). Overall, $S_{age}$ has a higher LUC-associated direct carbon flux than $S_{ageless}$ for the first 12 years,
and a lower one afterwards (Fig. 7a). The cross point for the cumulative LUC-associated direct fluxes
equal in $S_{age}$ and $S_{ageless}$ is around the $20^{th}$ year (Fig. 7b). When summing over the whole simulation period
(100 years), the cumulative fluxes by $S_{ageless}$ is lower in $S_{age}$ by about 11 kg C m$^{-2}$, or ~110 g C m$^{-2}$ yr$^{-1}$
(Fig. 7b) than $S_{ageless}$.
**3.1.3 LUC emission and its disaggregation into underlying component carbon fluxes**
As defined in Eq (4), the net LUC carbon emission ($E_{LUC}$) is diagnosed as the difference in NBP between
the LUC simulation and the control one. Since NBP is further a composite flux determined by carbon
uptake and releases (Eq. 3), the difference in $E_{LUC\ age}$ and $E_{LUC\ ageless}$ can be disaggregated into the effect of
each underlying flux, which differs between the LUC simulation and the control simulation. Figure 8
presents such disaggregation. All positive values indicate an enhanced carbon uptake or diminished
release in the LUC simulation compared to the control one, whereas negative values indicate the reverse
cases (i.e., negative values indicate a contribution to enhance $E_{LUC}$).

First of all, $S_{ageless}$ (no age dynamics) simulates a larger magnitude (i.e., a larger absolute $E_{LUC}$ value) of
mean annual $E_{LUC}$ than $S_{age}$ (with age dynamics), by about 26 g C m$^{-2}$ yr$^{-1}$. Second, for both simulations,
the simulated $E_{LUC}$ is an outcome of LUC-associated direct fluxes being compensated for by changes in
other fluxes, all of which have an effect to reduce $E_{LUC}$ in this example: NPP, heterotrophic respiration,
fire carbon emissions and agricultural harvest.

NPP is higher in LUC simulations than in the control. This is because young forests are established in the
former case (either by merging with existing forest patch or not), leading to a younger leaf age than in the
control simulation, which is parameterized to have a higher photosynthetic capacity than older leaves in
the model. This suggests the model can somewhat integrate the effect of recovering young forests or
intermediate-aged forests with a higher productivity than the old-growth forests, as reported by Tang et al.
(2014) using observation data.

Averaged over the LUC simulation period of 100 years, both $S_{age}$ and $S_{ageless}$ show lower heterotrophic
respiration ($F_{HR}$) than the control. This is because the biomass stock is lower in the LUC simulations





(despite a higher NPP, biomass turnover is accelerated due to site perturbation and wood collection in the
process of clearing forest for cropland), causing less litter input and less soil carbon stocks (data not
shown). The $S_{age}$ simulation shows a much smaller reduction of $F_{HR}$, mainly because a higher
belowground litter is maintained, which results from an abnormally high belowground litter input out of
land turnover, driven by a high belowground biomass, as explained in Sect. 3.1.1 (Fig. 6a).

Decreases in fire carbon emissions ($F_{Fire}$, from prognostically simulated 'natural fires' but not 'land-
clearing fires') in the LUC simulations in contrast with the control are because the aboveground litter
(dominant fuel for fires) is reduced by land turnover. Reductions in fire emissions, and reductions in
heterotrophic respiration, are thus driven by the same process, i.e., a reduction in aboveground standing
biomass. LUC simulations also result in lower agriculture harvest ($F_{AH}$, from cropland) although there is
no change in the cropland area; this is due to lower biomass in young crop, as the crop harvest is assumed
as a constant fraction of the biomass turnover  (i.e., routine mortality) at a daily time step. The lower crop
biomass in the LUC simulations here is because crop saplings are established on the first day of each
calendar year, right before the seasonal biomass peak for the southern hemisphere, which artificially
reduces the standing biomass.

Overall, the lower $E_{LUC}$ magnitude in $S_{age}$ is a result of the lower LUC-associated direct fluxes having
been partly compensated for by a higher heterotrophic respiration. The relative magnitudes between $E_{LUC}$
$_{age}$ and $E_{LUC\ ageless}$ are dominated by these two fluxes, while other fluxes play a less important role.
**3.2 Simulation over Southern Africa**
**3.2.1 Forest cohort area change as a result of historical land use change**
One of the useful features of our model development is to account for sub-grid forest age dynamics as a
result of historical land use change, as illustrated in Fig. 9 for Southern Africa. When no land use change
is included (S0, the control simulation), the areas of all forest cohorts are constant over time. Except that
younger cohorts have a very small area (<0.1 Mkm$^2$) (Cohort$_2$ and Cohort$_3$, probably due to improper
cohort thresholds on a very small number of grid cells), almost all forests are found in Cohort$_6$, which
represents mature forests. In S1 where only net land use change is considered, the area of Cohort$_6$
decreases consistently over time due to conversion of forest to other land cover types (Fig. 9a).
Occasional increases in areas of other younger cohorts are also present, corresponding to the periods
when forest gain happens due to net land use change, for instance, afforestation or reforestation around
1700s and in the latter half of the 20$^{th}$ century (Fig. 9a). This is consistent with our rule that forest from
abandonment of agriculture is established in the youngest cohort (Fig. 5b – on the right), and progressive



movement of forests from younger to older cohorts are also visible as the small waves in the curves of
Fig. 9b–f.

In the S2 simulation with both net land use change and land turnover, large areas of younger forests, in
particular of Cohort$_1$ and Cohort$_2$, begin to appear as a result of continual creation of forests from land
turnover, and subsequent moving of forests from Cohort$_1$ to Cohort$_2$. Their temporal changes over time
follow those of the forest area subject to land turnover, as shown in Fig. 9a (green dashed line). The area
of Cohort$_3$, however, does not see as much increase as in the two younger cohorts, because forests of
Cohort$_3$ are the primary target for clearance in land turnover and thus are incessantly converted back to
(shifting) agriculture. As a result, about half of mature forests (Cohort$_6$) are left intact from LUC by 2005
(Fig. 9h). Most interestingly, when there is a decline in the turnover-impacted area around 1700s (the
green arrow in Fig. 9a), a corresponding decline in the area of Cohort$_1$ is found because these forests
move into the next cohort. This pattern of decrease in the current cohort accompanied by the according
increase in the next one then propagates into other older cohorts with time, which results in a delayed
increase in Cohort$_5$ around 1750s (Fig. 9g), and finally in Cohort$_6$ as well (but less prominent because of
its already large area). This demonstrates the model feature of older forest recovery in case of decreased
land turnover or wood harvest, as explained in Fig. 5b (right hand side). Last, when we further include
forest harvest in S3 simulation, because wood harvest area only started to rise in the middle of 20$^{th}$
century, larger areas of Cohort$_1$ and Cohort$_2$ cohorts are found compared with S2 in the latter half of the
last century, and forest area in Cohort$_6$ is accordingly lower, being converted to younger cohorts as a
result of harvest.
**3.2.2 Cumulative LUC emissions**
Cumulative LUC emissions over 1501–2005 in the S$_{ageless}$ and S$_{age}$ simulations for Southern Africa are
shown in Fig. 10. In both simulations, including land turnover and wood harvest leads to higher total
LUC emissions, by roughly a factor of 1.5 in S$_{age}$ than in the S1 simulation with only net land use change,
and by a factor of 2 in S$_{ageless}$, respectively. Total carbon emissions from all LUC processes in S$_{age}$ are 14.2
Pg C, 35% lower than in S$_{ageless}$ (21.9 Pg C). The lower total LUC emissions in S$_{age}$ are mainly due to
lower emissions from land turnover, being 6.7 Pg C, almost half of those by S$_{ageless}$ (12.5 Pg C). This is
consistent with the findings of idealized grid cell simulation (Sect. 3.1.3).

Cumulative emissions from net land cover change (E$_{LUC net}$) diagnosed from Eq (4) are also lower in S$_{age}$
than in S$_{ageless}$ (6.2 versus 8.4 Pg C) (Fig. 10). This is mainly attributed to a few grid cells, where
occasional forest gains (i.e., afforestation or reforestation) occurred during some period over 1501–2005,
but eventually, all forests have been cleared. In such cases, occasional forest gains will lead to creation of



younger cohorts in the $S_{age}$ simulation; these younger cohorts have lower biomass carbon stock than the
otherwise mature forests in the $S_{ageless}$ simulation, hence leading to lower LUC emissions. As for wood
harvest, because the area subjected to harvest only started to increase around the middle of the 20$^{th}$
century (Fig. 9), in both $S_{age}$ and $S_{ageless}$ it is mainly mature forests or older cohorts that are harvested,
whose biomass density differ little (in the $S_{age}$ simulation, all secondary forests are locked in the
continuously expanding land turnover, so the forests subjected to harvest are taken first from older
cohorts). As a result, over the region of Southern Africa carbon emissions from wood harvest are almost
equal between the two simulations (Fig. 10).
**4 Discussion**
DGVMs, either used in an off-line mode or coupled with climate models, are powerful tools to investigate
the role of past and future land use change in the global carbon cycle perturbed by human activities
(Arneth et al., 2017; Le Quéré et al., 2016). Therefore, a more realistic representation of LUC processes
in these models is a scientific priority. We included two new features in ORCHIDEE-MICT: gross land
use change and forest wood harvest, and sub-grid vegetation cohorts. In a recent review (Prestele et al.,
2016), proper representation of gross land use change or sub-grid bi-directional land turnover has been
identified as one of the three major challenges in implementing LUC in DGVMs for credible climate
assessments. Large underestimation of LUC emissions would occur when gross land use change is
ignored, as shown by Wilkenskjeld et al. (2014), Stocker et al. (2014) and also by our results over
Southern Africa.

Shifting cultivation, or forest wood harvest, or more in general forest management, often involves a stable
fallow length or rotation cycle, which involves secondary forests rather than primary ones. In tropical
regions, fallow lengths in shifting cultivation range from 10 to 40 years (Bruun et al., 2006; Mertz et al.,
2008; Thrupp et al., 1997; van Vliet et al., 2012), with a tendency in reduction of fallow length. In Latin
American tropics, agricultural abandonment have already led to prominent growth of secondary forests
(Chazdon et al., 2016; Poorter et al., 2016). Forest management, including wood harvest, is more
common in temperate and boreal regions. In European forests, rotation lengths depend on tree species,
regional climate and management purposes (McGrath et al., 2015), ranging from 8–20 years in coppicing
systems in southern Europe to 80–120 years in northern countries. The prevalence of secondary forests
associated with land use and land use change therefore calls for their representation in DGVMs,
especially when modeling land use change. However, to our knowledge, integration of both land use
change and sub-grid secondary forests in DGVMs remains rarely reported. Yang et al. (2010) examined
the contribution of secondary forests to terrestrial carbon uptake using a vegetation model by explicitly



including secondary forest PFTs, but they did not include the dynamic clearing of secondary forests in
land use change, nor shifting cultivation. ORCHIDEE-CAN is especially designed to address forest
management and species change. Although some certain land use change is included there, but a full LUC
scheme addressing all possible LUC processes, including the gross change, is missing (Naudts et al.,

669    2015).


The gross land use change combined with sub-grid cohorts presented here has shown some promising
results. We first confirmed that including gross land use change leads to additional carbon emissions.
However, these additional emissions tend to be overestimated when secondary forests are not explicitly
accounted for. The idealized grid cell simulation well explained the mechanism driving such
overestimation in $S_{ageless}$ simulations at the regional scale. The forest aboveground biomass carbon stocks
subjected to LUC impacts, a large part of which are released to the atmosphere as instantaneous fluxes or
from later wood product degradation, are likely overestimated when secondary forests are absent in the
model. This has given rise to higher LUC emissions in $S_{ageless}$ simulations.

The results presented here are closely linked with our model parameterization and in particular, various
decision rules regarding which forest cohorts to apply for specific LUC processes. In order to examine the
influence by including gross land use change, we separated land use change into three LUC processes: net
land use change, land turnover and forest wood harvest. Land turnover and secondary forest harvest are
parameterized to target intermediate-aged cohorts as a priority. This is the core mechanism driving the
lower LUC emissions when sub-grid forest age structure is accounted for. As a preliminary effort to
demonstrate the model behaviour, the land turnover parameterization is heavily tied with the input LUC
forcing data (LUH1), so that the age of Cohort$_3$ (as the primary target for land turnover) is set as ~15
years, following the assumed mean residence time of shifting cultivation in LUH1 data set (Hurtt et al.,
2006). We admit that this parameterization is crucial, because it largely determines the rotation length in
the model, and consequently, the amount of carbon stocks subjected to LUC and the difference in
estimated LUC emissions between the two model configurations ($S_{age}$ and $S_{ageless}$). But in fact, because the
thresholds in woody mass to distinguish forest cohorts could be configured via a spatial map and such
maps could vary among different years, to apply temporally and spatially different turnover lengths is
rather straightforward in the model. Such feature is well considered in the model design and could be
tested given available forcing LUC data.

We now discuss some model features as our deliberate decisions and their potential influences in modeled
LUC impacts. First, the LUC module developed is intended for usage within DGVMs, and forced with



external data sets that provide information on land flows between different land cover types. It is not
intended to supersede a land use change model per se, which simulates land use change using other
available social and economic information such as population, food demand, wood demand, etc. (Hurtt et
al., 2016). In this sense, the LUC module implementation has to inevitably take into account the details of
information in forcing data that are available, and to reconcile the potential inconsistency between the
model and forcing data. For example, the LUC module presented here can accommodate forest wood
harvest from primary and secondary forests when these two sources are distinguished in the forcing data,
but hierarchical decision rules are also made when the model and forcing data disagrees (e.g., Fig. 5),
such as that prescribed "secondary forest wood harvest" can actually harvest a "primary forest" in the
model if all younger cohorts are exhausted.

Second, because of this clearly defined border of the LUC module to use land areas as the input
information, model output from OCHIDEE-MICT can potentially disagrees with the socio-economic
information used to generate the LUC forcing data. For instance, crop yield simulated by ORCHIDEE
may differ with that used to convert food demand/consumption to cropland area, so that simulated crop
output or food production will disagree with historical food demand in the real world. The same applies
on forestry wood production: simulated harvest wood volume might disagree with the wood volume
actually used to generate the harvest area information. This largely raises the issue that, to what extent the
information that drives land use change decisions can be *internally* integrated into DGVMs, for example,
to use directly crop production, rather than cropland area, or wood volume, rather than forest harvest area
as the model input. One potential obstacle is that statistical information (e.g., on wood volume demand) is
often available on regional basis (FAO global forest resource assessment, http://www.fao.org/forest-
resources-assessment/en/; eurostat, http://ec.europa.eu/eurostat/data/database), and complex decision rules
are needed to disintegrate such information on spatial grids that DGVMs are operated on. But in general,
there is need to streamline land use or land management decisions directly into DGVMs. ORCHIDEE-
CAN has integrated forest management decisions based on simulated tree diameters and stand density, so
that wood volume is actually an output from the model that can be validated against historical statistical
data (Naudts et al., 2015).

The developments presented here mainly build on a model structure to distinguish differently aged
cohorts. Nonetheless, we have built a better tool to address the impacts of historical land use change on
carbon cycle and climate with these developments. Forest demographic dynamics, which are shown to
have great impact on the current northern hemisphere carbon sink (Pan et al., 2011; Piao et al., 2009b),
either as a result of active afforestation, or agricultural abandonment or natural regeneration, could then





be explicitly investigated. The model also opens the possibility to verify modeled global and regional forest age distribution with that from either forest inventory or satellite imaging. On regional scale such as Europe, it is also possible to account for the LUC impact on full greenhouse gas balance, thanks to the recent developments in pasture module and cropland module (Chang et al., 2015; Wang et al., 2017).

## 5 Conclusions

We have presented new developments made in a global vegetation model, to include gross land use change and forest wood harvest, in combination with explicit representation of sub-grid forest age dynamics. The results are specific of the ORCHIDEE-MICT model, but the methods are generic for other DGVMs. We demonstrated that over Southern Africa, including gross land use change and forest harvest has led to additional carbon emissions compared to a case where only net transitions are included. However, these additional emissions are overestimated using the traditional approach where secondary forests are not accounted for in the model and quasi-primary forests are cleared for shifting cultivation (or land turnover). We therefore conclude that explicit inclusion of sub-grid secondary forests is crucial for more accurate estimation of land use change emissions. Our developments open the possibility to account for forest demography when evaluating LUC impacts on global carbon cycle and climate.

## 5 Code availability

The ORCHIDEE-MICT codes used here are a development version deposited on the SVN server: https://forge.ipsl.jussieu.fr/orchidee/browser/perso/chao.yue/ORCHIDEE-MICT-GLUC revision 4259 from the 20th April 2017. The code is open source, but readers interested in the model application are encouraged to contact the corresponding author.

## 6 Data availability

Primary data and scripts used in the analysis and other supplementary information that may be useful in reproducing the authors' work can be obtained by contacting the corresponding author.

## Acknowledgements

C. Yue and W. Li acknowledge the European Commission-funded project LUC4C (No. 603542). P. Ciais acknowledges the support from the European Research Council through Synergy grant ERC-2013-SyG-610028 "IMBALANCE-P".

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





**Figures and Tables**

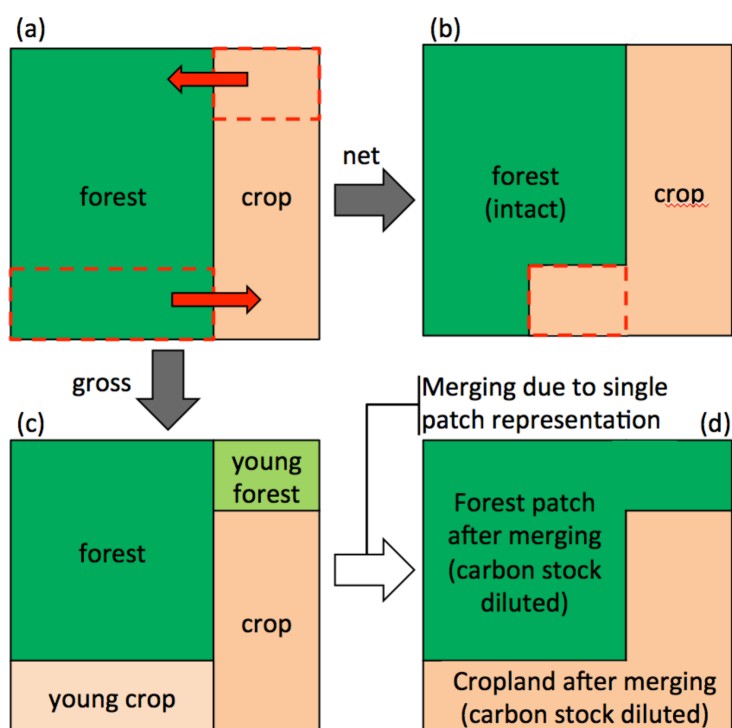


Fig. 1 Schematic illustration of gross versus net land use change, with each land cover type being
represented using a single patch within a model grid cell. The figure is adapted from Stocker et al. (2014).
(a) Original fractions of forest and cropland and land use transitions. Dashed red rectangles indicate areas
subject to LUC and red arrows indicate land flow direction. Here LUC consists of a net loss in forest and
a simultaneous bi-directional flow between forest and cropland. (b) Post-LUC fractions of forest and
cropland following the original LUC scheme of net transition only in ORCHIDEE. Bi-directional land
flow is omitted, with only cropland area being expanded to account for the net increase (as a result of the
net forest loss, as indicated by the dashed red rectangle). The soil carbon stock of the new cropland patch
is an area-weighted mean between that of the original cropland, and the legacy stock from the former
forest. Carbon stock of the remaining forest patch is left intact. (c) Intermediate post-LUC land cover
pattern after accounting for gross transition. Both the net loss of forest and bi-directional land flows are
accounted for, with two young patches of forest and cropland being established, respectively. (d) Final
state of post-LUC land cover after accounting for gross LUC with no sub-grid cohorts. The carbon stocks
of the remaining (original) forest and the newly created forest are immediately merged following LUC



because there are no sub-grid cohorts. The same applies for cropland as well. Note that although forest
and cropland fractions are ultimately the same as in (b), the carbon densities are different.



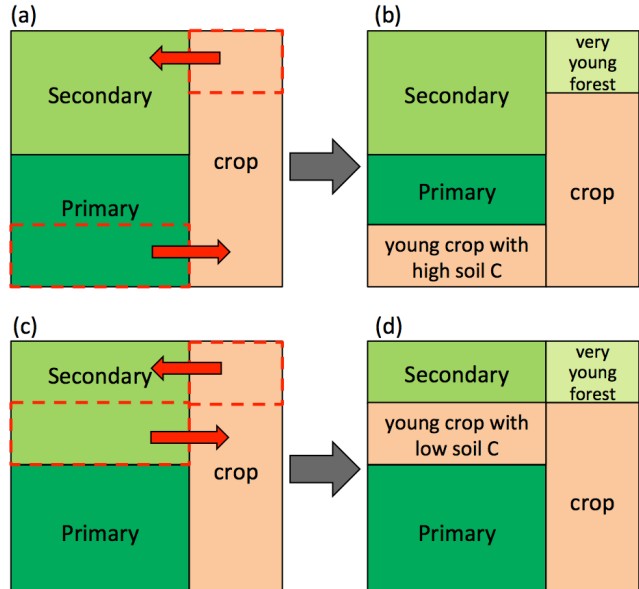


Fig. 2 Gross land use change involving forests with different ages, under a model scheme capable of
representing sub-grid vegetation cohorts. The figure is adapted from Stocker et al. (2014). LUC here is
similar as in Fig. 1, except that forest is no longer a single ageless patch but consists of two patches of
primary and secondary forests, i.e., having an age structure. (a) The same area of forest is converted to
cropland as in Fig. 1a but conversion is made from primary forest. (b) Consequently, a 'young' cropland
patch with rich legacy forest soil C is established. In the meanwhile, a very young forest patch is
established due to the bi-directional gross land flux. Because the model uses multiple sub-grid patches to
represent vegetation age structure (or differently aged cohorts), merging of patches with different carbon
stocks is no longer necessary. Subplot (c) shows an alternative to (a) where conversion of forest to
cropland is made on a secondary forest. Correspondingly, in subplot (d), which shows the post-LUC state
of (c), the established young cropland patch will have lower legacy soil C than that in (b).



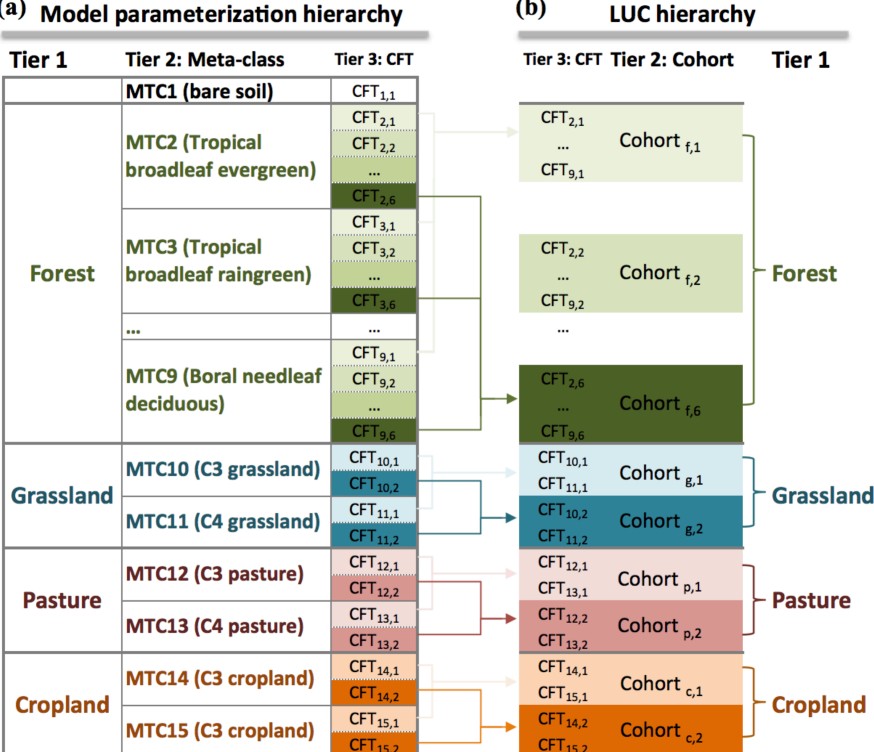


Fig. 3 Two parallel hierarchies from the model parameterization and land use change perspective. (a)
Sub-grid cohort function types (CFTs) as inheritances of meta-classes (MTCs) and the corresponding
parameterization hierarchy. There are in total 14 vegetative MTCs corresponding to four vegetation types.
The notation of $CFT_{i,j}$ indicates that it inherits from $MTC_i$ and belongs to the $j^{th}$ cohort ($Cohort_j$). Each
forest MTC has six cohorts, with $Cohort_1$ being the youngest and $Cohort_6$ the oldest, whereas each
herbaceous MTC is set tentatively to have two cohorts. Darker colors indicate older cohorts. (b) Within
the gross LUC module hierarchy, Tier 3 remains the level of CFT, but CFTs are re-organized to derive the
Tier 2 information based on the level of cohorts, under the same Tier 1 as in (a). A cohort baring the
notation of $Cohort_{v,i}$ indicates it belongs to vegetation type 'v' (where 'v' could be forest, natural
grassland, pasture and cropland) and meta-class 'i'. This re-organization of the hierarchy is to prepare for
properly allocating prescribed LUC transitions first onto the cohort level, then further to different CFTs
within each cohort.





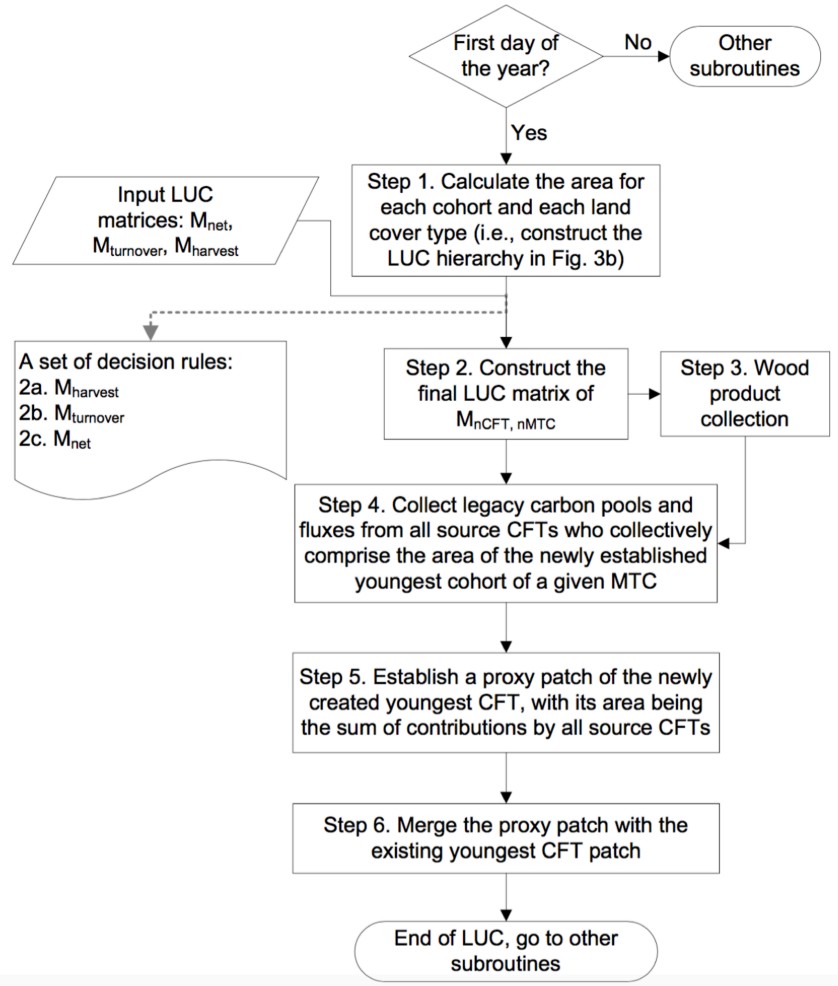


Fig. 4 Schematic representation of the new LUC scheme in ORCHIDEE-MICT accounting for net land
use change, land turnover and forest harvest in combination with sub-grid cohort representation.





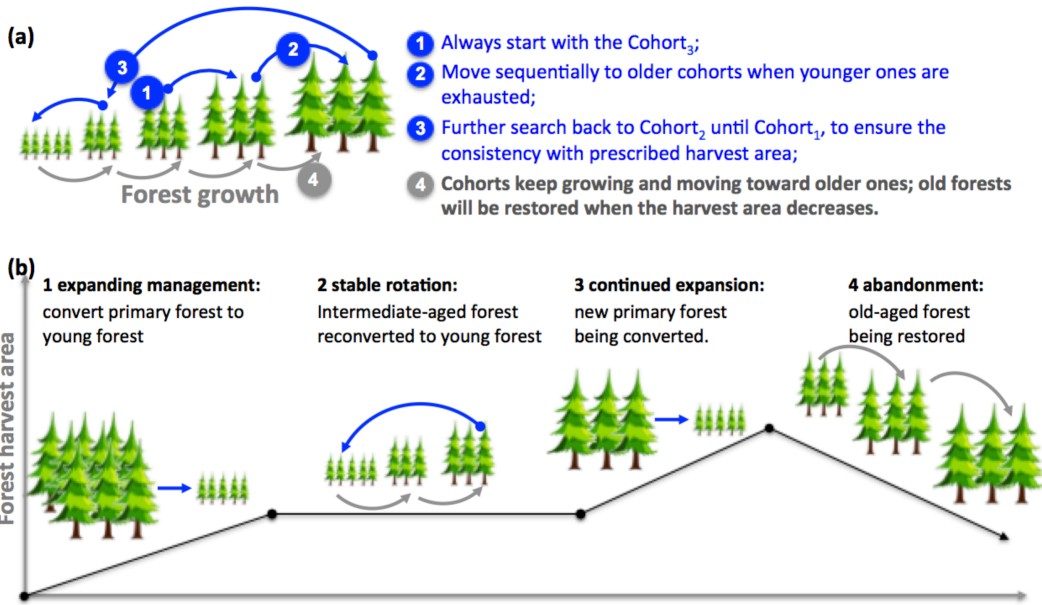


Fig. 5 Rules of selection of forest cohorts in the wood harvest to account for dynamic changes in the area
subjected to wood harvest over time. (a) Rule for the selection of forest cohort (blue arrows). Clear-cut
harvest (1) first starts with intermediate-aged cohort, then moves to older cohorts until the oldest one; (2)
if the prescribed harvested area still cannot be satisfied, then the selection will move back to the even
younger cohorts (3) until the youngest one until the prescribed harvested area is fulfilled. Independent of
the harvest activity is the movement of forests from younger cohorts to older ones because of growth
(gray arrows). (b) Example of cohort dynamics for changes in the harvest area over time shown in the
black curve: (1) before the onset of any harvest activity (i.e., after the model spin-up), only the oldest
cohorts are available so harvest starts with the primary forest; (2) for a stable harvest area, a steady-state
cycle is established involving only secondary forest (intermediate secondary cohorts being harvested
represented by the blue arrow, and younger cohorts growing represented by gray arrows); (3) then with an
increase in harvest area, more primary forests are harvested; (4) finally in this example, the harvest area
decreases, and older cohorts are restored.





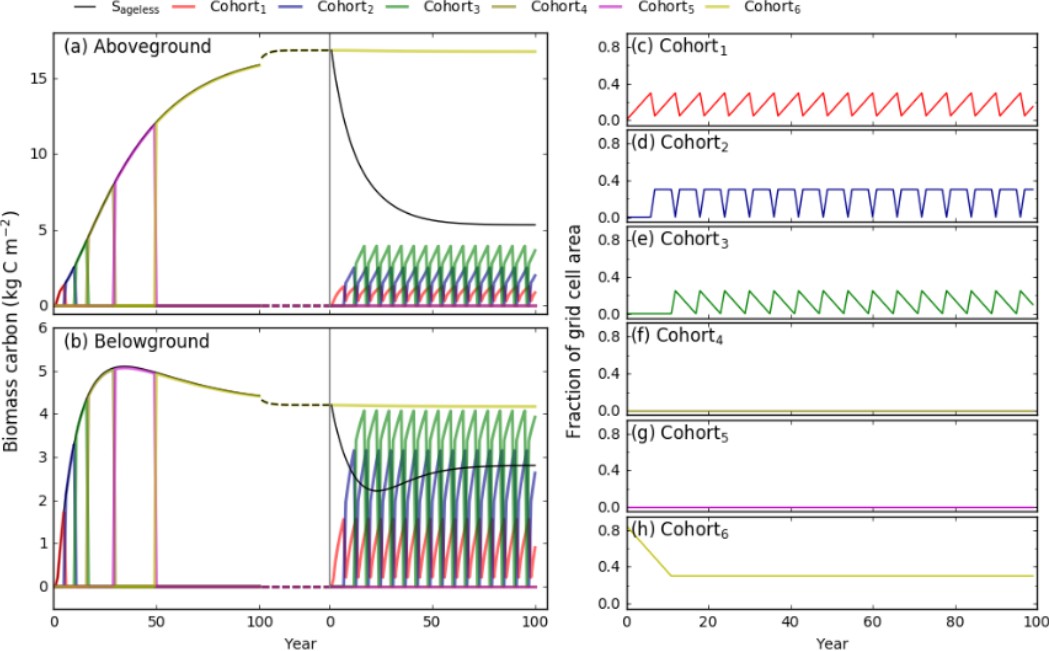


Fig. 6 Biomass carbon stock as simulated by two model configurations without ($S_{ageless}$) and with sub-grid
age dynamics for (a) aboveground biomass and (b) belowground biomass. Data shown are the biomass
accumulation during the spin-up simulation (which lasts for 450 years, from Year 0 until the end of
dashed line) and transient simulation (which lasts for 100 years) where an annual forest-cropland turnover
with 5% of the grid cell area is applied. Vertical gray lines indicate the end of the spin-up and start of
transient simulations. Subplot (c)–(h) show ground coverage by different forest cohorts as fractions of
grid cell during the transient simulation only.





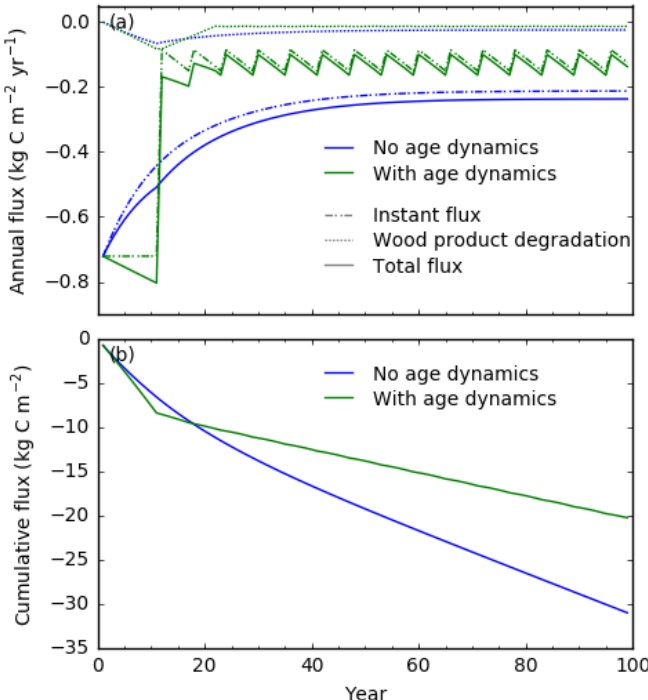


Fig. 7 (a) Carbon fluxes directly associated with LUC (negative values for carbon lost from ecosystems):
instantaneous flux (dash-dotted line), flux from wood products degradation (dotted line) and the total flux
(solid line) for simulations with (green) and without (blue) sub-grid age dynamics. (b) Cumulative LUC-
associated direct fluxes (the sum of instantaneous and wood products degradation fluxes) for simulations
with (green) and without (blue) sub-grid age dynamics. Data are shown for an annual forest-cropland
turnover of 5% of the grid cell area for 100 years.





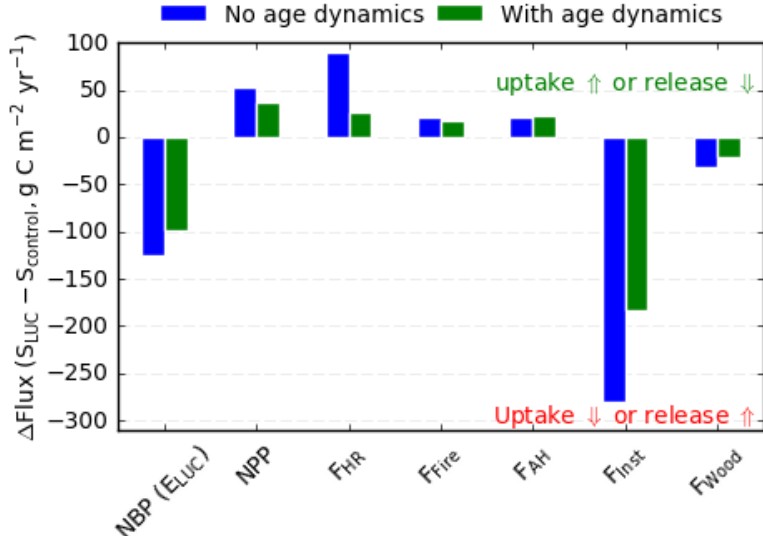


Fig. 8 Mean annual carbon flux differences between the LUC and control simulations over 100 years for
an annual forest-cropland turnover with 5% of the grid cell area for two model configurations: without
(blue) and with sub-grid age dynamics (green). Positive (negative) values indicate contributions to
enhanced carbon sink (source) in LUC simulation compared to the control one, either by stronger
(weaker) carbon uptake or smaller (stronger) carbon release. $E_{LUC}$ is shown as a negative value here, i.e.,
the LUC simulation has a lower NBP than the control one, indicating an effect of net carbon source by
LUC.

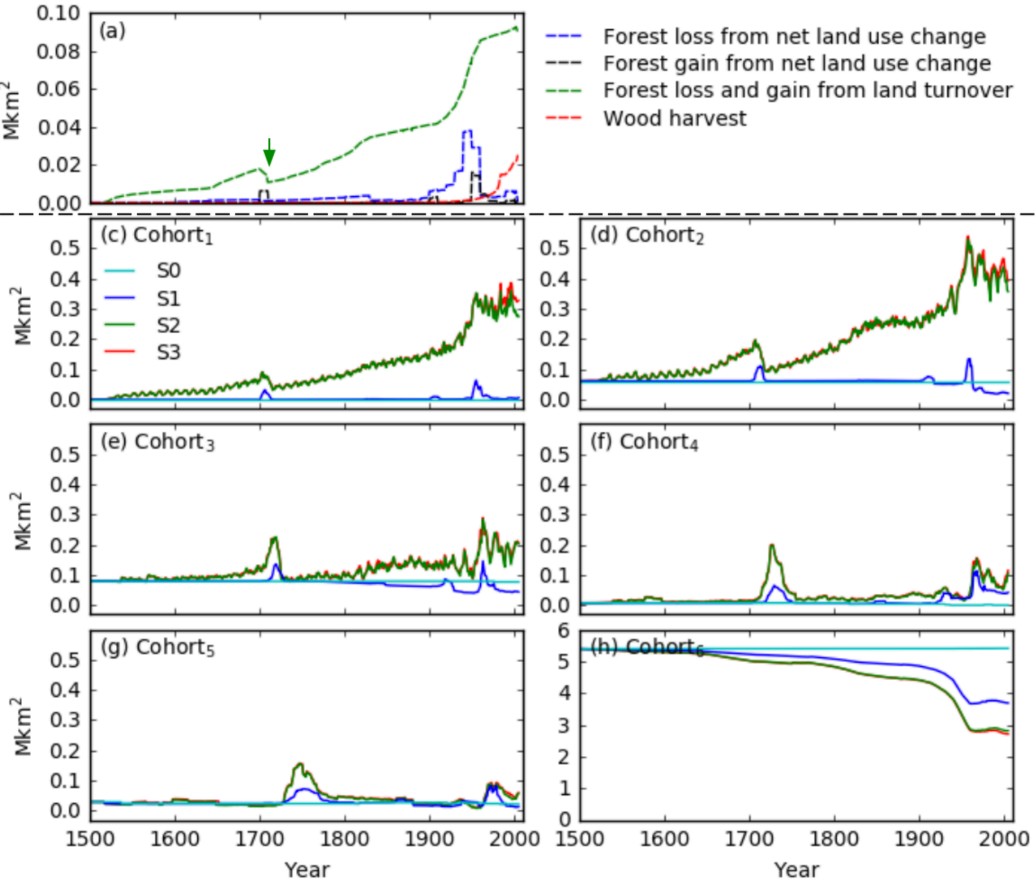


Fig. 9 Areas subject to historical land use change and the resulting modeled temporal changes in areas of
different forest cohorts in Southern Africa. (a) Areas subjected to historical land use change in which
forests are involved. Data are from LUH1 reconstruction (Hurtt et al., 2011) after adaption for
ORCHIDEE-MICT. Three types of LUC activities are shown and their effects elucidated by factorial
simulations (Table 2). These are: forest loss (blue dashed line) and gain (black dashed line) resulting from
net land use change, forest involved in land turnover (both loss and gain in equal amount, green dashed
line), and forest area subjected to wood harvest (red dashed line). (b)–(h) Areas of forest cohorts (Cohort$_1$
= the youngest, Cohort$_6$ = the oldest) for four factorial simulations (Table 2) where no land use change
occurs in S0, and the three LUC types are added in a factorial set-up in S1 (net land use change, blue solid
line), S2 (net land use change + land turnover, green solid line) and S3 (net land use change + land
turnover + wood harvest, red solid line). Note y-scale values in subplot (a) and (h) differ from others.





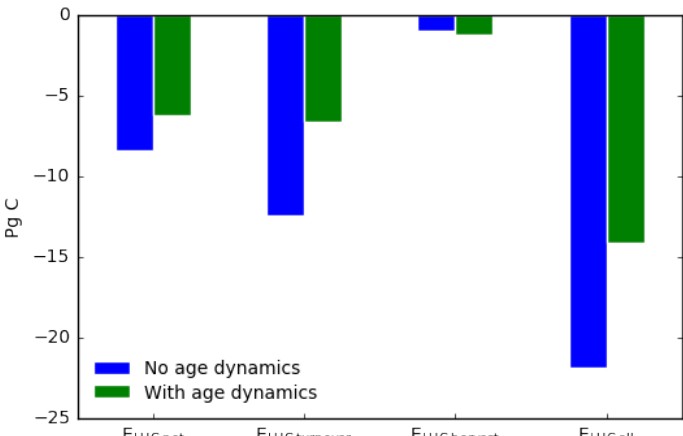


Fig. 10 Cumulative carbon emissions over 1501-2005 from land use change over the region of South
Africa from three LUC processes: net land use change ($E_{LUC\ net}$), land turnover ($E_{LUC\ turnover}$) and wood
harvest ($E_{LUC\ harvest}$) and the sum of them ($E_{LUC\ all}$), for two model configurations: with ($S_{age}$, green color)
and without ($S_{ageless}$, blue color) sub-grid age dynamics.


Table 1. Fractions of aboveground woody biomass lost immediately to the atmosphere during a forest
clearing, and channeled to 10-year and 100-year turnover wood product pools. These fractions are
different depending on forest biomes.

|  | Tropical forest | Temperate forest | Boreal forest |
|---|---|---|---|
| $F_{instant}$ | 0.897 | 0.597 | 0.597 |
| $F_{10yr}$ | 0.103 | 0.299 | 0.299 |
| $F_{100yr}$ | 0 | 0.104 | 0.104 |




Table 2 Factorial simulations to separate contributions from each of the three LUC processes: net land use
change ($E_{LUC\ net}$), land turnover ($E_{LUC\ turnover}$) and wood harvest ($E_{LUC\ harvest}$). The plus signs ("+") indicate
that the corresponding processes (matrices) are included in the simulations, with $S0_{age}$ ($S0_{ageless}$) having no
LUC activities to $S3_{age}$ ($S3_{ageless}$) including all LUC processes. The land use carbon emissions are
quantified as the difference in NBP between simulations with and without LUC or a specific LUC process
(Eq. 4).



Simulations and LUC processes included

| Simulations | Net land use change | Land turnover | Wood harvest |
|---|---|---|---|
| $S0_{age}$, $S0_{ageless}$ | | | |
| $S1_{age}$, $S1_{ageless}$ | + | | |
| $S2_{age}$, $S2_{ageless}$ | + | + | |
| $S3_{age}$, $S3_{ageless}$ | + | + | + |

Diagnostic of LUC emissions

| Without age dynamics | With age dynamics |
|---|---|
| $E_{LUC\ net,\ ageless} = NBP_{S1ageless} - NBP_{S0ageless}$, | $E_{LUC\ net,\ age} = NBP_{S1age} - NBP_{S0age}$ |
| $E_{LUC\ turnover,\ ageless} = NBP_{S2ageless} - NBP_{S1ageless}$, | $E_{LUC\ turnover,\ age} = NBP_{S2age} - NBP_{S1age}$, |
| $E_{LUC\ harvest,\ ageless} = NBP_{S3ageless} - NBP_{S2ageless}$ | $E_{LUC\ harvest,\ age} = NBP_{S3age} - NBP_{S2age}$, |
