# Peer review of "Representing anthropogenic gross land use change, wood harvest and forest 1 2 age dynamics in a global vegetation model ORCHIDEE-MICT v8.4.2 3 Chao Yue1, Philippe Ciais1, Sebastiaan Luyssaert2, Wei Li1, Matthew J. McGrath1, Jinfeng Chang<su"

_Geoscientific Model Development, 2017_

## Short Comment (SC1) · 21 Jul 2017

The link https://forge.ipsl.jussieu.fr/orchidee/browser/branches/ORCHIDEE-MICT nominated as the access to the code in the manuscript does not work. It seems to require an login. This needs to be fixed. It is also a bit usual to use an SVN version number to refer to a model version. I would like to suggest that the authors tag the particular revision in SVN and use this tag as a reference instead. Obviously releases of ORCHIDEE have been tagged before although the last release seems to be a few years old. Nevertheless I am strongly encouraging the authors to do this and to

change the title accordingly. In order to guarantee persistent access to the release the use of a DOI is strongly encouraged but not enforced. It is also recommended to add a brief statement on license of use. If possible we also strongly encourage authors to provide data and scripts as supplements for the manuscript again in order to guarantee persistent access to the information.
* * *

---

## Referee Comment (RC1) · Anonymous Referee #1 · 1 Aug 2017

**General Comment:**

This study touches on the issue of the representation of shifting cultivation in the dynamic vegetation model. The new model features including a better description on PFTs (plant function types) demography, wood harvest and shifting cultivation at a subgrid scale. The behavior of the enhanced model was tested both at a small scale and at a regional scale over an old growth forest (Miombo/dry woodlands) in South Africa. The model result shows that the new development has a robust representation of shifting cultivation during a long-term simulation period and the carbon emission due to

the land use change has been underestimated without the consideration of gross land use change (including shifting cultivation, age class PFT and wood harvest). The most important term for this net emission is contributed from the biomass burning due to shifting cultivation activities (the $F_{Inst}$ term in Eq. (3)). The manuscript was written in a good shape with a detail model description and its experimental design, and the new model feature opens the opportunities for the scientific community to study the research issue such as the effects of shifting cultivation between different biomes on the climate from different soil types and climate zones.

**Specific Comment:**
I suggest the authors to provide a more detailed description and adequate reference of each term in the Eq. (3), which are the crucial parts of mathematical representation for the bio-physical/chemical processes. For example, the "$F_{HR}$" term is often parameterised as function of surface temperature, and it also could be parameterised as function both of surface temperature and soil moisture (Chang et al. 2008). In the view of result presented by the authors, "$F_{Inst}$" term is the major source of the net CO2 emission from the shifting cultivation between forests and croplands. I would also like to understand the sensitivity of this term to the state variables, such as soil temperature, soil carbon stock and ect. in the model.

In this paragraph (P8L241-L245), I was confused about the description of the recruitment in a forest. Does the natural recruitment in a forest increase the original forest cover fraction (Diluted the carbon stock)? Or, the forest cover fraction is always fixed and the recruitment only increases the carbon stock.

The author choose a dry woodland as an example to demonstrate the model behavior of shifting cultivation at a dry and warm climate zone. Regarding to the design of the land surface model (ORCHIDEE) is for a large scale study, I think it would be able to apply this new feature for a tropical peat land forest and the model behavior should be also welcome and interesting for the readers in the Earth System Modeling community.

Reference:
Chang, S.-C., K.-H. Tseng, Y.-J. Hsia, C.-P. Wang, and J.-T. Wu. 2008. Soil respiration in a subtropical montane cloud forest in Taiwan. Agric. Forest Meteorol. 148: 788-798

**Technical Comment:**
P2L59: the definition of "M" $10^6(million) or 10^9(mega)$?

P2L65: reference of "Hasis et al. 2015" is missing the reference list

P4L110: Some recent developments..., please cite more references

P5L158: ..."Fig 1d"... to ..."Fig. 1d"...

P8L239: ...are properly defined. Please explain how to define the criteria for the cohort thresholds.

P9L279: the index i, j have been already used. It should be replaced by another indices, such as k, l. P13L395, L404: The description of $F_{Fire}$ for Eq. (3) is missed.

P13L414: ..."simulations and Le Quere et al. (2016)"... I suggest to rephrase it to ..."simulations and the existing global carbon budget dataset (Le Quere et al., 2016)".

P15L473-L474: six CFTs but only five ages (3, 9, 15, 30, 50) in the text

P15L481: the reason for choosing 65

P19L599: The Fig. 9 sub-index for "b" can't find the Figure 9. Please revise it for the consistence between the context and figure.

P21L667-L669: Please give an example for the possible missing process in the land use change.

P22L702: The citation of "Hurtt et al. 2016" is not in the reference list.

P22L711: Typo: ...O"R"CHIDEE-MICT...

P22L723: ..."is need to streamline land use"... This is a bad English structure. I would

recommend to rephrase it as . . ."is needed to streamlining to land use". . .

P23L734-L736: It is a sentence with a bad English structure. Please rephrase it.

P33L989: Add a line for "$S_{age}$" simulation. I was confused about the zero cover fraction for both Cohort4 and Cohort5. For a 100 year simulation, the Cohort4 and Cohort5 supposed to have dynamic changes in the cover fraction. Pease explain the zero cover fraction for Cohort4 and Cohort5 in the main text.

P36L1014: Please check the label of the Fig.9. sub-label "b" is missed.
* * *

---

## Referee Comment (RC2) · B. Stocker (Referee) · 18 Sep 2017

The paper by Yue et al. describes the implementation of gross land use change within the ORCHIDEE Dynamic Global Vegetation Model. This implementation relies on an explicit and separate treatment of six different age cohorts of land "patches". C dynamics are simulated separately within each patch and cohort age priority for conversion from forest to agricultural land is specified explicitly. It is shown both at the level of an individual gridcell and at the regional scale (Southern Africa) that this leads to lower LUC-related $CO_2$ emissions compared to a simulation where age cohorts are not distinguished in a simulation that accounts for gross land use change.

This is a substantial and very complex step in model development and improves the realism of simulations of the anthropogenic land use change. The paper provides a detailed and in some parts rather technical and model-speciffic description of the implementation. It convincingly shows for a single example gridcell how biomass is simulated to accumulate and transition through cohorts of different age and how it reaches a dynamic steady state under constant gross land use change regime (no expansion, constant land turnover). In that sense, one can conclude that the model works - arguably the most important statement of this paper.

The authors then go on to investigate the effect of gross versus net land use change and the effect of separating six age cohorts (versus averaging all into a single age cohort) for land use change CO2 emissions of southern Africa. They conclude that "emissions from bi-directional land turnover alone are 35% lower in Sage than Sageless. (abstract)" and that the effect of gross versus net is to increase emissions by a factor of 2 (for "S_ageless") and 1.5 (for "S_age"). I have some concerns regarding the presentation of these conclusions, and regarding the scope (investigating age cohort effects) itself. One more (major) issue is regarding model spin up (see further below).

As stated by the authors (l.87-90), the present paper is not the first one to implement a model for simulating gross land use transitions. Stocker et al. (2014) and Wilkenskjeld et al. (2014) are cited. However, the authors forgot to refer to Shevliakova et al. (2009), GBC, who also implemented multiple age cohorts for simulating gross land use change. It should also be made clear that at least Shevliakova et al. (2009) and Stocker et al. (2014) (not Wilkenskjeld, as far as I am aware) did make a distinction between at least two age cohorts. Referring to "traditional approaches where a single patch is used for a given land cover type" (abstract, l.26) and presenting results of the simulation "S_ageless" as representative for "traditional approaches" is thus a bit misleading.

The present paper was submitted on 14 May 2017. On 26 July 2017, Yue, Ciais

and Li submitted a paper to Biogeosciences Discussions (https://www.biogeosciences-discuss.net/bg-2017-329/), where the same model is applied to investigate essentially the same questions, but this time at the global scale. The regional focus of the present paper on southern Africa may appear arbitrary at first, but makes sense. Apparently, authors preferred to devote a full paper to model description and evaluation and a second full paper to a global application. In my view, this is a viable way to go and the large work that went into developing this model warrants two separate papers. However, I find the delineation of their respective scope a bit unsatisfying. Readers will likely be left asking themselves why authors didn't present results from global simulations in the present (GMDD) paper - a relatively small additional step in terms of additional work. Simultaneously, readers of the BGD paper might be left wondering what the additional insight of that paper is after already the GMDD paper concluded that accounting for separate age cohorts reduces the effect of gross versus net LUC emissions.

A solution for that is to reinforce the value of the present (GMDD) paper in terms of its model documentation and dissemination aspects. Section 2.1.4. is very technical and might be too specific for the ORCHIDEE model, limiting its value for a wider readership. Code is not made publicly accessible (only upon request) and the study is therefore not reproducible. However, authors note the "clearly defined border of the LUC module". In my view, it would be highly beneficial for the present paper to provide open access, reproducible code along with the paper. The module itself should be able to be decoupled from the rest of ORCHIDEE and some "synthetic" simulations should be possible, where land use transitions and cohorts dynamics are simulated by published parts of code. (I don't understand why this is not strictly required anyway for GMDD.)

In any case, I encourage that the authors find a solution to finding a better delineation between their parallel submissions currently under review here and in BGD.

Regarding model spin up: Fig. 6 shows that if a constant land turnover rate is applied during the transient simulation, but not during spinup, biomass C stocks attain the "wrong" equilibrium. I.e. stocks decline after being subjected to continuous land

turnover to a new steady state, reached after around 50 years (under a tropical climate). Soil C stocks likely take longer to attain a new steady state and in cold climates even more so. If simulations are evaluated from the start of the transient simulation, then land-atmosphere C fluxes related to reaching this new steady state confound results. How is this treated when, for example, doing a historical simulation starting in 1850? Shouldn't a continuous land turnover pattern be applied already during spin up in order to avoid these disequilibrium fluxes?

MINOR:

* l.61: . . . emissions *of $CO_2$* (Houghton et al., 1999) . . .

* l. 67 "Given the importance of historical LUC emissions and its large uncertainty, a more realistic representation of LUC processes and land management in DGVMs is desirable". Improving realism rarely reduces uncertainty ($\sim$model spread).

* The net-versus-gross LUC question is introduced only on l.73 - in my view too late. Preceeding paragraphs detract attention from the questions at hand here.

* As pointed out on l.95, accounting for gross land use change is relevant more generally for appropriately simulating sub-grid scale bi-directional land use transitions and is not only relevant in shifting cultivation agriculture. The term shifting cultivation refers to a specific form of small-holder agriculture and doesn't encompass all sub-grid scale bi-directional land use transitions. I am aware that the previous literature on modelling these effects used the terms shifting cultivation and gross land use change (or land turnover) more or less interchangeably. Maybe worth stating here (in introduction) what shifting cultivation actually is (see Heinimann et al., 2017, PLOSOne: https://doi.org/10.1371/journal.pone.0184479).

* l.105: reference for "dilution approach"?

* l.122 "r3247": Better use SVN tags than SVN version numbers for reference.

* l. 277 (Eq.1):

* Fig. 6: Very nice plot! Would be very informative to have a curve for total biomass across all cohorts in the simulation with age distinction (to make it comparable to the black curve for S_ageless).

* Fig. 7: Is this figure referenced in the text? Where?

---

## Short Comment (SC2) · 18 Sep 2017

The manuscript by Yue and colleagues presents model development in ORCHIDEE-MICT, incorporating a forest age structure and gross land use transitions, including shifting cultivation. Both aspects were subject to several papers in the recent years and it would be helpful if the authors set their implementation and their findings more (and more accurately) in context of the already published literature.

In particular:

[Figure]

1. Regarding the effects of net vs. gross transitions, there has been a recent multi-model study by Arneth et al. (2017) that showed the importance of tree harvesting and land clearing from shifting cultivation. In this paper seven models were used to determine the influence of wood harvest and shifting cultivation. It might be helpful to relate to the findings of Arneth et al. (2017).

2. The described approach to model gross transitions as matrices looks very similar to the implementation of gross transitions in the DGVM JSBACH, as described by Reick et al. (2013), which has not been mentioned at all in the manuscript so far (please also see the comment on the lines 87-93 below). It would be helpful to include some comparisons of the way Yue et al. represent gross transitions and the way it is presented in Reick et al. (2013). The same might hold for the mentioned paper describing LPX-Bern (Stocker et al., 2014). There are two further models listed in the 2015 update on the global carbon budget that include gross transitions (Table 5; Le Quere et al., 2015): CLM4.5 (Oleson et al., 2013) and Visit (Kato et al., 2013), which might also be worth looking at.

3. There are several DGVMs that have some kind of age structure, e.g. LPJ-Guess with its gap dynamics (Smith et al., 2014) and LM3V (Shevliakova et al., 2009). The latter is particularly interesting for the manuscript of Yue et al. because of the combination of simulated secondary regrowth and land use and land management, including shifting cultivation.

4. I do not understand which of the implementations regarding age structure stem from ORCHIDEE-CAN and which are newly developed in this study (l. 190-221), and I think it would be helpful if the authors could revisit this paragraph for clarity. Particularly, I do not understand how cohorts are ageing in ORCHIDEE-MICT. Since this might be a critical aspect for the described carbon dynamics it would be helpful if the authors could put some more emphasis in describing the ageing of the forest, maybe an additional Figure could help.

Other commments

lines 87-93: This paragraph is unfortunately not correct. Gross transitions are implemented in the DGVM JSBACH (see Reick et al., 2013), not in an emulator. Also, Wilkenskjeld et al. (2014) did not use an emulator but the carbon cycle sub-module of JSBACH, for efficient comparisons of net and gross transitions. Furthermore, JSBACH with gross transitions has already been used in the MPIESM simulations for CMIP5 and in TRENDYv4 simulations used in the global carbon budget in 2015 (Le Quere et al., 2015). In this budget, two further models beside JSBACH did include gross transitions (see "shifting cultivation", Table 5, Le Quere et al., 2015). The reason why no model included gross transitions in the 2016 update of the global carbon budget was because the LUH2v2h data set was not ready: "The more comprehensive harmonised land-use data set (Hurtt et al., 2011), which also includes fractional data on primary vegetation and secondary vegetation, as well as all underlying transitions between land-use states, has not been made available yet for this year. Hence, the reduced ensemble of DGVMs that can simulate the LUC flux from the HYDE data set only." (Le Quere et al., 2016).

line 115: "sub-grid sub-grid"

line 113: "plant function types" -> plant functional types

line 137: "forgings" -> forcings

lines 215-217: this assumption might not be correct for natural grasslands and pastures (see e.g. Nyawira et al. 2016 and references therein).

line 285: "The cohort age subject to LUC of is one..." -> remove the of

line 328: According to their webpage (http://gsweb1vh2.umd.edu/luh_data/LUHa.v1/readme.txt) LUH1 also makes a distinction of harvest from mature and young forest. Do you use this information in your model, too? Furthermore, LUH contains "harvest from non-forested land", is this information used?

line 341: "first go first for" -> first go for

line 347: should this maybe be secondary?

line 359: "to ensure the their" -> to ensure that their?

line 386: but it respires in the grid cell where it is harvested?

line 403: I do not understand this sentence

line 427: remove the "and"?

line 430: replace "on" with "by"?

line 445: held constant or held as constants

line 447: a hypothetical scenario

line 448: I do not understand the sentence "Forest harvest of the same intensity..."

lines 556-561: But why is the NPP in simulations with age dynamics smaller? Is the forest in these simulations not yet as productive than intermediate-age forest?

line 702: Do you mean Hurtt et al. 2006? Else the reference is missing.

lines 710-715: It might be helpful to mention here again that LUH does include biomass harvest but that this is not used in your model.

line 748 this is section 6

line 753 and this section 7

line 1015: Fig. 9 does not include a "panel b"

References

Arneth et al.: Historical carbon dioxide emissions caused by land-use changes are possibly larger than assumed, Nat. Geosci., 10(2), 79–84, doi:10.1038/ngeo2882, 2017.

Kato, E., Kinoshita, T., Ito, A., Kawamiya, M., and Yamagata, Y.: Evaluation of spatially explicit emission scenario of land-use change and biomass burning using a process-based biogeochemical model, Journal of Land Use Science, 8, 104–122, 2013.

Le Quere et al.: Global Carbon Budget 2015, Earth Syst. Sci. Data, 7, 349–396, doi:10.5194/essd-7-349-2015, 2015.

Le Quere et al.: Global Carbon Budget 2016, Earth Syst. Sci. Data, 8, 605–649, doi:10.5194/essd-7-349-2015, 2016.

Oleson et al.: Technical Description of version 4.5 of the Community Land Model (CLM), National Center for Atmospheric Research, Boulder, CO, USA, available at: http://www.cesm.ucar.edu/models/cesm1.2/clm/CLM45_Tech_Note.pdf (last access: 8 November 2016), 2013.

Nyawira S. S., Nabel J. E. M. S., Don A., Brovkin V. & Pongratz J.: Soil carbon response to land-use change: evaluation of a global vegetation model using observational meta-analyses, Biogeosciences 13(19), 5661–5675, 2016.

Reick, C. H., Raddatz, T., Brovkin, V., and Gayler, V.: The representation of natural and anthropogenic land cover change in MPI-ESM, Journal of Advances in Modeling Earth Systems, 5, 459–482, 2013.

Shevliakova, E., S. W. Pacala, S. Malyshev, G. C. Hurtt, P. C. D. Milly, J. P. Caspersen, L. T. Sentman, J. P. Fisk, C. Wirth, and C. Crevoisier: Carbon cycling under 300 years of land use change: Importance of the secondary vegetation sink, Global Biogeochem. Cycles, 23, GB2022, doi:10.1029/2007GB003176, 2009.

Smith et al.: Implications of incorporating N cycling and N limitations on primary production in an individual-based dynamic vegetation model. Biogeosciences 11, 2027–2054, 2014.

Stocker, B. D., Feissli, F., Strassmann, K. M., Spahni, R. & Joos, F. Past and future carbon fluxes from land use change, shifting cultivation and wood harvest. Tellus B 66,

23188, 2014.

Wilkenskjeld, S., Kloster, S., Pongratz, J., Raddatz, T. and Reick, C. H.: Comparing the influence of net 914 and gross anthropogenic land-use and land-cover changes on the carbon cycle in the MPI-ESM, 915 Biogeosciences, 11(17), 4817–4828, doi:10.5194/bg-11-4817-2014, 2014.

---

## Author Comment (AC1) · 25 Nov 2017

Following the suggestion, we have tagged this version of code as ORCHIDEE-MICT v8.4.2 and will change the title of revised manuscript accordingly. As the model presented here is part of ORCHIDEE-MICT, the issues of code availability and license generally follow the discussion paper by Guimberteau et al. 2017 gmd-2017-122 on ORCHIDEE-MICT and our responses generally follow theirs (https://www.geosci-model-dev-discuss.net/gmd-2017-122/gmd-2017-122-AC1-supplement.pdf). Regarding the code availability and license issue, we have added in the section of "Code availability" the following sentences: "The source code for ORCHIDEE-MICT version 8.4.2 is available online (https://forge.ipsl.jussieu.fr/orchidee/browser/branches/ORCHIDEE-MICT/tags/ORCHIDEE_MICT_GLUC_8.4.2.) but its access is restricted to registered users. Request can ben sent to the corresponding author for a username and password for code access. ORCHIDEE-MICT is governed by the CeCILL license under French law and abiding by the rules of distribution of free software. One can use, modify and/or redistribute the software under the terms of the CeCILL license as circulated by CEA, CNRS and INRIA at the following URL: http://www.cecill.info." Regarding data availability, we have stated in the "Data availability" section: "Primary data and scripts used in the analysis and other supplementary information that may be useful in reproducing the authors' work can be obtained by contacting the corresponding author".

---

## Author Comment (AC2) · 25 Nov 2017

**General Comment:**

This study touches on the issue of the representation of shifting cultivation in the dynamic vegetation model. The new model features including a better description on PFTs (plant function types) demography, wood harvest and shifting cultivation at a sub-grid scale. The behavior of the enhanced model was tested both at a small scale and at a regional scale over an old growth forest (Miombo/dry woodlands) in South Africa. The model result shows that the new development has a robust representation of shifting cultivation during a long-term simulation period and the carbon emission due to the land use change has been underestimated without the consideration of gross land use change (including shifting cultivation, age class PFT and wood harvest). The most important term for this net emission is contributed from the biomass burning due to shifting cultivation activities (the $F_{Inst}$ term in Eq. (3)). The manuscript was written in a good shape with a detail model description and its experimental design, and the new model feature opens the opportunities for the scientific community to study the research issue such as the effects of shifting cultivation between different biomes on the climate from different soil types and climate zones.

We appreciate the reviewer's efforts to review our paper. Please see our point-to-point response as below. All the revised texts in response to the reviewer's request are tracked in the updated manuscript.

**Specific Comment:**

I suggest the authors to provide a more detailed description and adequate reference of each term in the Eq. (3), which are the crucial parts of mathematical representation for the biophysical/chemical processes. For example, the "$F_{HR}$" term is often parameterised as function of surface temperature, and it also could be parameterised as function both of surface temperature and soil moisture (Chang et al. 2008). In the view of result presented by the authors, "$F_{Inst}$" term is the major source of the net CO2 emission from the shifting cultivation between forests and croplands. I would also like to understand the sensitivity of this term to the state variables, such as soil temperature, soil carbon stock and ect. in the model.

Reference:Chang, S.-C., K.-H. Tseng, Y.-J. Hsia, C.-P. Wang, and J.-T. Wu. 2008. Soil respiration in a subtropical montane cloud forest in Taiwan. Agric. Forest Meteorol. 148: 788-798

All the terms in Eq. (3) are now explained in more details. Further references are provided when necessary. $F_{Inst}$ represents the instant carbon fluxes to the atmosphere in forest clearing and is determined in the model on an annual time scale. It depends only on the wood mass of the forests being cleared and not directly on soil status including the temperature and moisture. This is now explained clearly in the revised texts. We added the following texts in Sect. 2.1.5: *"Carbon in the two wood product pools is then released into the atmosphere according to their respective turnover times, and this flux contributes to the overall land carbon balance as a source term (see the next section)."*, *"Agricultural harvest and associated fluxes to the atmosphere through food consumption or livestock feeding are assumed to happen locally in the model, without considering spatial relocation by international trade."*. We added the following texts in Sect. 2.2.1: *"$F_{Inst}$ and $F_{Wood}$ are both fluxes on an annual time scale that depend only on wood mass at the time of forest clearing and the respective wood product degradation rates (see Sect. 2.1.5). FHR is simulated at a time step of 30 minutes and depend on soil temperature and moisture. $F_{Fire}$ is simulated with a prognostic fire module SPITFIRE (Yue et al., 2015)."*

In this paragraph (P8L241-L245), I was confused about the description of the recruitment in a forest. Does the natural recruitment in a forest increase the original forest cover fraction (Diluted the carbon stock)? Or, the forest cover fraction is always fixed and the recruitment only increases the carbon stock.

We apologize for this confusion in the original text. The focus here is to describe how forest cover fractions are handled in the process of natural mortality and recruitment, as our paper focuses on land cover change representation in the model. Natural recruitment from regeneration in a forest does not increase the original forest cover fraction. It does not either dilute the existing carbon stock (here, the original texts are inaccurate in its description). Instead, recruitment increases individual density and renews part of leaves (by updating leaf age composition in the model). The recruited sapling biomass is incorporated into the existing biomass only when the latter is virtually zero while a larger-than-zero ground fraction is prescribed. We revised the relevant texts in the paper as below and hope it is clearer (Sect. 2.1.3):

*Natural forest mortality in ORCHIDEE could be either prescribed as a constant rate or dynamically simulated, but mortality takes effects by reducing the amount of existing biomass only, with the coverage of the concerned forest patch being unchanged. Likewise, recruitment increases forest individual density and update leaf age composition and other relevant variables, but again, forest coverage remains unchanged. These features are necessary, as the original ORCHIDEE model does not take into account forest demography. As explained in Krinner et al. (2015, page 8), recruitment sapling biomass is only incorporated when the existing biomasses is virtually zero while a larger-than-zero ground coverage is prescribed. These features remain the same in the case of with sub-grid cohorts, i.e., forest mortality or natural recruitment does not modify forest cohort ground coverage. In addition, forest mortality and subsequent regeneration due to forest fires are handled in a similar manner.*

The author choose a dry woodland as an example to demonstrate the model behavior of shifting cultivation at a dry and warm climate zone. Regarding to the design of the land surface model (ORCHIDEE) is for a large scale study, I think it would be able to apply this new feature for a tropical peat land forest and the model behavior should be also welcome and interesting for the readers in the Earth System Modeling community.

We agree with the reviewer that forest clearing in tropical peat land forest can be an interesting case to apply our model. In a companion paper (https://www.biogeosciences-discuss.net/bg-2017-329/) where we apply our model to investigate historical land-use change carbon emissions from shifting cultivation, there are some shifting cultivation activities in tropical Asia being included. However, the hydrological impacts on carbon due to land use change on peat-land forest must have not been adequately represented mainly because peat-land-related hydrological process and soil processes are not represented in the model version used here. There is a parallel model development in ORCHIDEE aiming for including peat land process (https://www.geosci-model-dev-discuss.net/gmd-2017-155/). In the future, these developments could be integrated for a more sensible representation of peat land-related land use change.

**Technical Comment:**

P2L59: the definition of "M" $10^6$(million) or$10^9$(mega)?

It means $10^6$ (million). This is indicated in the revised manuscript.

P2L65: reference of "Hansis et al. 2015" is missing the reference list

Done.

P4L110: Some recent developments. . ., please cite more references

In response to the comments by other reviewers as well, we have added an overview table of DGVMs having implemented gross land use change, and more references are added in the introduction section.

P5L158: ..."Fig 1d"... to ..."Fig. 1d"...

Done.

P8L239: . . .are properly defined. Please explain how to define the criteria for the cohort thresholds.

This has been explained in detail in Sect. 2.2.3 in the original manuscript. To not increase the manuscript length by making repeats, the section 2.2.3 is now cited in the Sect. 2.1.3.

P9L279: the index i, j have been already used. It should be replaced by another indices, such as k, l.

We argue that it is convenient and an implicitly agreed practice to denote an element of a matrix M as $M_{ij}$. In our case we suppose readers can easily distinguish that here the indices i,j are different from the ones used before in Sect. 2.1.3. So this notation is maintained.

P13L395, L404: The description of $F_{Fire}$ for Eq. (3) is missed.

It is explained in the revised texts, in response as well to the first specific comment by the reviewer.

P13L414: ..."simulations and Le Quere et al. (2016)"... I suggest to rephrase it to . . ."simulations and the existing global carbon budget dataset (Le Quere et al., 2016)".

Done.

P15L473-L474: six CFTs but only five ages (3, 9, 15, 30, 50) in the text

The last cohort (Cohort$_6$) corresponds to the mature or primary forest and therefore its age (i.e., years) is not given as an exact number. To remove the potential confusion, we denote the age of Cohort$_6$ as >50 years in the revised manuscript.

P15L481: the reason for choosing 65%.

This value here is chosen tentatively and more for a demonstration purpose. The key point is to separate agricultural lands (croplands and pastures) into two broad age groups assuming that they have different soil carbon stocks. In general, because changes of soil carbon stock following land use change are spatially highly diverse and depend on many factors including the land cover types before and after the transition, the model feature described here is more for informative demonstrating purpose rather than having solid scientific significance. This is primarily due to the fact that soil moisture is simulated in the model on the basis of water columns, and soil temperature over the whole grid cell rather than cohorts, as explained in the text (Sect. 2.2.3, 2$^{nd}$ paragraph). To fully track the soil carbon trajectory after land use change, a much larger number of cohorts for herbaceous vegetation are needed, but this is limited by the computing power when running simulation over the globe. Overall, this feature is more like a "place holder" whose function needs to be explored and parameterization has to be improved in the future model application. We inserted at the end of Sect. 2.2.3 the following sentences to clarify this: "*Overall, this feature of separating herbaceous MTCs into multiple cohorts is coded more as a "place holder" for the current stage of model development rather than having solid scientific significance. To fully track soil carbon stocks of different vegetation types and their transient changes following land use change, a much larger number of cohorts are needed. But for a global application, this is limited by the computation efficiency.*"

P19L599: The Fig. 9 sub-index for "b" can't find the Figure 9. Please revise it for the consistence between the context and figure.

Fig. 9 is now revised.

P21L667-L669: Please give an example for the possible missing process in the land use change.

The example is given in the original text, e.g., gross land use change.

P22L702: The citation of "Hurtt et al. 2016" is not in the reference list.

This is corrected.

P22L711: Typo: . . .O"R"CHIDEE-MICT. . .

Thanks for pointing this out. We apologize for this typo. It has been corrected.

P22L723: . . ."is need to streamline land use". . . This is a bad English structure. I would recommend to rephrase it as . . ."is needed to streamlining to land use". . .

*We change this sentence to: But in general and over a long term, land use or land management decisions need to be integrated directly into DGVMs.*

P23L734-L736: It is a sentence with a bad English structure. Please rephrase it.

*We modified these two sentences to make them more concrete: These developments also make it possible to verify modeled global and regional forest age distribution using independent age information from either forest inventory or remote sensing. The model version used here has incorporated the developments in pasture and cropland modules (Chang et al., 2015; Wang et al., 2017). On a regional scale such as Europe, where the comprehensive forcing data are available, it is possible to go beyond the carbon emissions only by LUC activities, but also to include LUC-induced changes in emissions of other greenhouse gases such as methane and nitrogen oxide.*

P33L989: Add a line for "$S_{age}$" simulation. I was confused about the zero cover fraction for both Cohort4 and Cohort5. For a 100 year simulation, the Cohort4 and Cohort5 supposed to have dynamic changes in the cover fraction. Pease explain the zero cover fraction for Cohort4 and Cohort5 in the main text.

The $S_{age}$ simulation is shown as each individual cohort from $Cohort_1$ to $Cohort_6$. This is now explained more clearly in the revised figure caption. As this figure shows a simulation of an annual forest-cropland turnover of 5% of grid cell area and the clearing of forest targets primarily on $Cohort_3$, this cohort has been converted to cropland before having the chance to move to $Cohort_4$. This explains the zero fractions of $Cohort_4$ and $Cohort_5$. This point is also explained in the revised text.

P36L1014: Please check the label of the Fig.9. sub-label "b" is missed.

This has been corrected.

---

## Author Comment (AC3) · 25 Nov 2017

The paper by Yue et al. describes the implementation of gross land use change within the ORCHIDEE Dynamic Global Vegetation Model. This implementation relies on an explicit and separate treatment of six different age cohorts of land "patches". C dynamics are simulated separately within each patch and cohort age priority for conversion from forest to agricultural land is specified explicitly. It is shown both at the level of an individual gridcell and at the regional scale (Southern Africa) that this leads to lower LUC-related $CO_2$ emissions compared to a simulation where age cohorts are not distinguished in a simulation that accounts for gross land use change.

This is a substantial and very complex step in model development and improves the realism of simulations of the anthropogenic land use change. The paper provides a detailed and in some parts rather technical and model-specific description of the implementation. It convincingly shows for a single example gridcell how biomass is simulated to accumulate and transition through cohorts of different age and how it reaches a dynamic steady state under constant gross land use change regime (no expansion, constant land turnover). In that sense, one can conclude that the model works - arguably the most important statement of this paper.

[R1] We appreciate the reviewer's efforts to review our paper and thanks for the general positive comments. The most model-specific section is probably Sect. 2.1.3, where cohort implementation has been described in detail in ORCHIDEE-MICT. This is necessary for understanding other sections. Furthermore, this could also give insights to other similar DGVMs (e.g., JSBACH, CLM) to implement similar schemes. Sec. 2.1.4, as we can argue, might seem model-specific at the first sight but actually is not — because two key model features, i.e., the necessity to introduce a priority decision rule and allocation of LUC-impacted cohort on different underlying vegetation types, can be needed as well when other DGVMs will try to implement LUC processes with vegetation demography. Thus the development presented here can be potentially useful for other similar DGVMs. We take the chance of addressing the reviewer's comments in the paragraph below to cite relevant studies and make close comparisons when describing our model development, to make the model descriptions more relevant for other DGVMs.

The authors then go on to investigate the effect of gross versus net land use change and the effect of separating six age cohorts (versus averaging all into a single age cohort) for land use change $CO_2$ emissions of southern Africa. They conclude that "emissions from bi-directional land turnover alone are 35% lower in Sage than Sageless. (abstract)" and that the effect of gross versus net is to increase emissions by a factor of 2 (for "S_ageless") and 1.5 (for "S_age"). I have some

concerns regarding the presentation of these conclusions, and regarding the scope (investigating age cohort effects) itself. One more (major) issue is regarding model spin up (see further below).

As stated by the authors (l.87-90), the present paper is not the first one to implement a model for simulating gross land use transitions. Stocker et al. (2014) and Wilkenskjeld et al. (2014) are cited. However, the authors forgot to refer to Shevliakova et al. (2009), GBC, who also implemented multiple age cohorts for simulating gross land use change. It should also be made clear that at least Shevliakova et al. (2009) and Stocker et al. (2014) (not Wilkenskjeld, as far as I am aware) did make a distinction between at least two age cohorts. Referring to "traditional approaches where a single patch is used for a given land cover type" (abstract, l.26) and presenting results of the simulation "S_ageless" as representative for "traditional approaches" is thus a bit misleading.

[R2] Thanks for the reviewer pointing out the work of Shevliakova et al. (2009) and Stocker et al. (2014). Such expression of "traditional approaches" is now removed in the texts. The model implementations of Shevliakova et al. (2009) and Stocker et al. (2014) are now discussed closely with our implementations in the revised text where relevant.

The present paper was submitted on 14 May 2017. On 26 July 2017, Yue, Ciais and Li submitted a paper to Biogeosciences Discussions (https://www.biogeosciences- discuss.net/bg-2017-329/), where the same model is applied to investigate essentially the same questions, but this time at the global scale. The regional focus of the present paper on southern Africa may appear arbitrary at first, but makes sense. Apparently, authors preferred to devote a full paper to model description and evaluation and a second full paper to a global application. In my view, this is a viable way to go and the large work that went into developing this model warrants two separate papers. However, I find the delineation of their respective scope a bit unsatisfying. Readers will likely be left asking themselves why authors didn't present results from global simulations in the present (GMDD) paper - a relatively small additional step in terms of additional work. Simultaneously, readers of the BGD paper might be left wondering what the additional insight of that paper is after already the GMDD paper concluded that accounting for separate age cohorts reduces the effect of gross versus net LUC emissions.

[R3] We greatly appreciate the reviewer's efforts to review both our papers and the holistic approach to the reviewing process. The separation of the two papers, and the arrangement of the contents for each of them, are based on several considerations: (1) It will be very lengthy to include both model developments and application in a single paper, so we decide to separate the work into two papers, with one focusing on model development description and exemplifying its application, and the other one focusing on the global application and implications for quantifying historical LUC emissions. We appreciate that the reviewer agreed on this approach. (2) The inclusion of forest demography and related cohorts is a key feature of the current paper. On top of this we implemented in the model a series of priority rules on which forest cohort to be targeted based on different LUC processes (Fig. 5). To exemplify the setting of forest cohort boundaries, and the impact of the implemented priority rule on forest demography dynamics (Fig. 9 in the original manuscript), a concrete example beyond the idealized single grid cell simulation is needed. This is the major motivation to include the southern Africa case study. (3) We argue that whether the results from the Southern Africa simulation belong to "illustrating the model behaviour" or to "scientific results" can be discussed. We tend to include Fig. 9 to show the

model behaviour as it is closely linked to the priority decision rules in LUC as explained in detail in 2.1.4. While in the bg-2017-329 paper, focus is given to the resulting LUC carbon emissions, their spatial and regional patterns and relevant comparisons with other studies. So the results in bg-2017-329 are not just a small step as argued by the reviewer, although the central message of that paper is in line with what's found by the idealized site-scale simulation in the current paper. These arguments/motivations are now included briefly in the revised texts in the introduction section.

A solution for that is to reinforce the value of the present (GMDD) paper in terms of its model documentation and dissemination aspects. Section 2.1.4. is very technical and might be too specific for the ORCHIDEE model, limiting its value for a wider readership. Code is not made publicly accessible (only upon request) and the study is therefore not reproducible. However, authors note the "clearly defined border of the LUC module". In my view, it would be highly beneficial for the present paper to provide open access, reproducible code along with the paper. The module itself should be able to be decoupled from the rest of ORCHIDEE and some "synthetic" simulations should be possible, where land use transitions and cohorts dynamics are simulated by published parts of code. (I don't understand why this is not strictly required anyway for GMDD.)

[R4] As we described in the response to the reviewers' first comment, Sect. 2.1.4 is necessary to understand the model behaviour and we believe it can also provide insights for other DGVMs. The model codes are in principle open source but according to the policy of the lab, the access is limited to registered users. A username and password will be given upon contact on the corresponding author in order to access the code. On the other hand, the developed module is complex, so interested readers are encouraged to contact the corresponding author to get some navigation through the codes and facilitate their understanding. We are confident that after some adaptation, the codes can be migrated into other models. However, it is challenging to design the codes as fully independent, isolated and pluggable into other DGVMs readily, because to the least extent, the codes are intended to work in the ORCHIDEE code environment.

In any case, I encourage that the authors find a solution to finding a better delineation between their parallel submissions currently under review here and in BGD.

[R5] Based on our responses above (R3), and in view of the reviewer's comments on our parallel bg-2017-329 paper, we revised both papers to make a clearer delineation in their scopes: (1) scopes are clearly defined in each introduction, with the current GMD paper focusing on model documentation and examination / illustration of model behaviour and the BG-paper focusing on model application at global scale with a focus on the 20th Century. (2) Fig. 10 in the original manuscript, along with relevant method sections have been removed in the revised GMD paper, with Fig. 9 being kept focusing on cohort dynamics resulting from historical land use change that are highly relevant to our implemented cohort priority rules in the model. (3) Model documentation is enhanced, with dissemination aspects being strengthened. In particular, DGVMs having already implemented gross land use change have been referred to and discussed in parallel with our implementation where relevant in the revised manuscript, in response to several reviewers' comments on this aspect. (4) We performed a series of additional simulations to investigate the sensitivity of derived land turnover emissions to the targeted cohort age and biomass in the African continent, to investigate the impact of delineating different cohorts on the

simulated land use emissions. But this result is included in the BG-paper as we think it's more relevant there. (5) In the BG-paper, the implication of our finding, i.e., lower emissions when taking into account age structure, is further discussed in relevance with our model implementation and the work of Arneth et al. (2017).

Regarding model spin up: Fig. 6 shows that if a constant land turnover rate is applied during the transient simulation, but not during spinup, biomass C stocks attain the "wrong" equilibrium. I.e. stocks decline after being subjected to continuous land turnover to a new steady state, reached after around 50 years (under a tropical climate). Soil C stocks likely take longer to attain a new steady state and in cold climates even more so. If simulations are evaluated from the start of the transient simulation, then land-atmosphere C fluxes related to reaching this new steady state confound results. How is this treated when, for example, doing a historical simulation starting in 1850? Shouldn't a continuous land turnover pattern be applied already during spin up in order to avoid these disequilibrium fluxes?

[R6] We agree with the reviewer that ideally some form of land turnover should be included in the spin-up runs. Likewise, natural stand-replacing disturbances should be included as well. However, unfortunately, neither of them is included in our current simulation. This point is acknowledged in the revised BG-paper where LUC emissions are examined in more details. On the other hand, as we start our simulation from Year 1501 in BG paper, LUC emissions from 1850 are unlikely impacted by a lack of land turnover in the spin-up and the estimations can still hold. This latter point is also discussed in the revised BG-paper.

MINOR:

* l.61: . . . emissions *of CO2* (Houghton et al., 1999) . . .

Done.

* l. 67 "Given the importance of historical LUC emissions and its large uncertainty, a more realistic representation of LUC processes and land management in DGVMs is desirable". Improving realism rarely reduces uncertainty (~model spread).

We agree on this. We changed the original sentence to "*Given the importance of historical LUC emissions and the necessity for their better understanding in order for a better idea of the future land-based mitigation potential, a more realistic representation of LUC processes and land management in DGVMs is desirable.*"

* The net-versus-gross LUC question is introduced only on l.73 - in my view too late. Preceeding paragraphs detract attention from the questions at hand here.

We have shortened the 1st and 2nd paragraph. We believe the lengths of these two paragraphs after revision provide a minimum scientific background of land-use change emissions before diving into the more technical description of gross versus net land use change.

* As pointed out on l.95, accounting for gross land use change is relevant more generally for appropriately simulating sub-grid scale bi-directional land use transitions and is not only relevant

in shifting cultivation agriculture. The term shifting cultivation refers to a specific form of smallholder agriculture and doesn't encompass all sub-grid scale bi-directional land use transitions. I am aware that the previous literature on modelling these effects used the terms shifting cultivation and gross land use change (or land turnover) more or less interchangeably. Maybe worth stating here (in introduction) what shifting cultivation actually is (see Heinimann et al., 2017, PLOSOne: https://doi.org/10.1371/journal.pone.0184479).

We inserted a brief definition of shifting cultivation in this paragraph following the suggestion by the reviewer. Heinimann et al. 2017 is cited as well. The inserted texts are: "*A typical example is shifting cultivation, a form of smallholder subsistence agriculture primarily occurring in tropical regions that involves clearing a forest for a non-permanent cropland, which is often abandoned later. Shifting cultivation was historically important in many tropical regions for the subsistence of indigenous people (Hurtt et al., 2006; Lanly, 1985) although more recently it has been in the process of being superseded by more intensified land management (Heinimann et al., 2017).*"

* l.105: reference for "dilution approach"?*

This term occurs in the ORCHIDEE documentation but cannot be found in literature. Thus it is removed from the revised texts.

l.122 "r3247": Better use SVN tags than SVN version numbers for reference.

Following the reviewer's suggestion, the code has been tagged as ORCHIDEE-MICT v8.4.2. The title is also changed accordingly.

* l. 277 (Eq.1):

The reviewer may have some comments here but it seems they did not show up.

* Fig. 6: Very nice plot! Would be very informative to have a curve for total biomass across all cohorts in the simulation with age distinction (to make it comparable to the black curve for S_ageless).

Biomass carbon stocks for all cohorts with age distinction, and for the single forest patch in the $S_{ageless}$ simulation are all based on per unit area of forest. The different cohorts in $S_{age}$ simulation are spatially distinct or separated so it does not make sense to add them together.

* Fig. 7: Is this figure referenced in the text? Where?

Yes. Fig. 7a and 7b are discussed in section 3.1.2 in the text.

---

## Author Comment (AC4) · 25 Nov 2017

The manuscript by Yue and colleagues presents model development in ORCHIDEE-MICT, incorporating a forest age structure and gross land use transitions, including shifting cultivation. Both aspects were subject to several papers in the recent years and it would be helpful if the authors set their implementation and their findings more (and more accurately) in context of the already published literature.

We thank the reviewer for the efforts to review our paper. As suggested, we included an overview table of the gross land use change implementations in current DGVMs. The introduction, model description and discussion sections are revised to take into account the existing work, and to make our implementation more relevant with other DGVMs. In addition, the whole manuscript has been carefully checked to avoid minor editing errors such as those raised by the reviewer in the technical comments.

In particular:

1. Regarding the effects of net vs. gross transitions, there has been a recent multimodel study by Arneth et al. (2017) that showed the importance of tree harvesting and land clearing from shifting cultivation. In this paper seven models were used to determine the influence of wood harvest and shifting cultivation. It might be helpful to relate to the findings of Arneth et al. (2017).

The work or Arneth et al. (2017) has been included in the discussion in our paper. Their work is now cited in the introduction as well in the revised manuscript. In both sections of introduction and discussion, references to this work are expanded.

2. The described approach to model gross transitions as matrices looks very similar to the implementation of gross transitions in the DGVM JSBACH, as described by Reick et al. (2013), which has not been mentioned at all in the manuscript so far (please also see the comment on the lines 87-93 below). It would be helpful to include some comparisons of the way Yue et al. represent gross transitions and the way it is presented in Reick et al. (2013). The same might hold for the mentioned paper describing LPX-Bern (Stocker et al., 2014). There are two further models listed in the 2015 update on the global carbon budget that include gross transitions (Table 5; Le Quere et al., 2015): CLM4.5 (Oleson et al., 2013) and Visit (Kato et al., 2013), which might also be worth looking at.

The similarity between Reick et al. (2013) and our work is that both are based on LUHv1 data and include gross transitions. There are two major differences: (1) Reich et al. (2013) focused on

reconciling dynamic vegetation process and external forcing data, while in our paper the dynamic vegetation is turned off and we relied on reconstructed historical land cover time series that are made consistent with the model. (2) Reick et al. (2013) made it an internal JSBACH model decision process on how to convert LUHv1 land cover types (i.e., primary and secondary natural lands) into model plant functional types (i.e., forest versus grassland, the pasture rule etc.). Whereas we focus on including different aged land cohorts in the ORCHIDEE model and implementing a set of hierarchical rules regarding the land cohorts subjected to different land use change processes. The allocation of natural lands into forest versus grasslands, and the reconciliation of LUHv1 land cover distributions and the current-day satellite-based PFT map, instead, are handled independently by external preparations of reconstructed historical PFT map time series. These differences are now discussed in the revised text where they're relevant. We also included references to other models as mentioned by the reviewer. In response to Beni Stocker's comments on a companion paper of the current study (bg-2017-39, Biogeosciences Discussions), we added an overview table of current DGVMs with implementations of gross land use change. For CLM4.5, we contacted Peter Lawrence and Danica Lombardozzi (personal communications) and they confirm that gross land use change has not yet been included in CLM4.5 but will be included in CLM5.

3. There are several DGVMs that have some kind of age structure, e.g. LPJ-Guess with its gap dynamics (Smith et al., 2014) and LM3V (Shevliakova et al., 2009). The latter is particularly interesting for the manuscript of Yue et al. because of the combination of simulated secondary regrowth and land use and land management, including shifting cultivation.

We thank the reviewer for referring to these highly relevant works. They're now cited and discussed in the revised paper in the introduction and discussion sections.

4. I do not understand which of the implementations regarding age structure stem from ORCHIDEE-CAN and which are newly developed in this study (l. 190-221), and I think it would be helpful if the authors could revisit this paragraph for clarity. Particularly, I do not understand how cohorts are ageing in ORCHIDEE-MICT. Since this might be a critical aspect for the described carbon dynamics it would be helpful if the authors could put some more emphasis in describing the ageing of the forest, maybe an additional Figure could help.

The basic approach and code base to introducing sub-grid cohorts are brought from previous developments in ORCHIDEE-CAN, which was made with the purpose to represent sub-grid forests of different age classes. This is stated in the original manuscript (line 205–206). To make this clearer, we inserted the following sentence after the original line 206: "*The code base to include sub-grid forest cohorts are migrated from ORCHIDEE-CAN, with substantial adaptions being made in ORCHIDEE-MICT. Except for this, all other LUC developments have been achieved within the current study.*" As ORCHIDEE-MICT is based on a single-leaf model, the aging of cohorts is simply represented by moving the concerned cohort to the next (older) one when its wood mass exceeds the cohort upper boundary. Except for cohort boundary, no further cohort-specific parameterization is done, so essentially all cohorts are governed by the same set of biophysical and ecological parameter values. However, in ORCHIDEE-MICT there do exist some "aging" processes to approximate some key changes when a forest ages, notably, the NPP allocation to belowground sapwood decreases with the time since establishment, that is, more biomass is allocated belowground to develop roots for young trees. We inserted the following

sentences at the end of the 3ʳᵈ paragraph in Sect. 2.1.3 to clarify this: "*Forest ages by moving from the current cohort to the next one when the woody biomass exceeds the cohort upper boundary. Except for the cohort boundaries, no further cohort-specific parameterizations are done, so essentially all cohorts are governed by the same set of biophysical and ecological parameter values. However, in ORCHIDEE-MICT there are indeed some simple "aging" processes to proximate the key changes when a forest ages, notably, the NPP allocation to belowground sapwood decreases with the time since establishment.*" We don't think an additional figure is needed so it is not provided.

lines 87-93: This paragraph is unfortunately not correct. Gross transitions are implemented in the DGVM JSBACH (see Reick et al., 2013), not in an emulator. Also, Wilkenskjeld et al. (2014) did not use an emulator but the carbon cycle sub-module of JSBACH, for efficient comparisons of net and gross transitions. Furthermore, JSBACH with gross transitions has already been used in the MPIESM simulations for CMIP5 and in TRENDYv4 simulations used in the global carbon budget in 2015 (Le Quere et al., 2015). In this budget, two further models beside JSBACH did include gross transitions (see "shifting cultivation", Table 5, Le Quere et al., 2015). The reason why no model included gross transitions in the 2016 update of the global carbon budget was because the LUH2v2h data set was not ready: "The more comprehensive harmonised land-use data set (Hurtt et al., 2011), which also includes fractional data on primary vegetation and secondary vegetation, as well as all underlying transitions between land-use states, has not been made available yet for this year. Hence, the reduced ensemble of DGVMs that can simulate the LUC flux from the HYDE data set only." (Le Quere et al., 2016).

Thanks for the reviewer for pointing out this mistake. It is now corrected. We added an overview table for DGVMs that have implemented gross land use changes (Table 1 in the revised manuscript).

line 115: "sub-grid sub-grid"

done.

line 113: "plant function types" -> plant functional types

done.

line 137: "forgings" -> forcings

done.

lines 215-217: this assumption might not be correct for natural grasslands and pastures (see e.g. Nyawira et al. 2016 and references therein).

We agree with the reviewer that our parameterization of herbaceous MTCs in terms of soil carbon changes cannot accommodate changes of SOC in all different LUC types. The effectiveness of this feature of differentiating herbaceous MTCs is limited by the model's simulation of soil temperature and moisture and the computation efficiency (as is explained in the original text, line 481–486). Therefore, this feature is more for informative purpose and serves as

a "place holder" for the future improvement in this scheme, rather than having solid scientific significance. We added in the revised two blocks of texts to clarify these points. We inserted the following texts in the 4[th] paragraph of Sect. 2.1.2: "*Because the directional change of soil carbon largely depends on the vegetation types before and after LUC and climate conditions (Don et al., 2011; Poeplau et al., 2011), ideally agricultural cohorts from different origins should be differentiated. However, to avoid exploding the total number of cohorts and the associated computation demand, as a first attempt, we simply divide each herbaceous MTC into two broad sub-grid cohorts according to their soil carbon stocks and without considering their origins. We expect that such a parameterization can accommodate some typical LUC processes, such as the conversion of forest to cropland where soil carbon usually decreases with time, but not all LUC types (for instance, soil carbon stock increases when a forest is converted to a pasture).*". We inserted in the last paragraph of Sect. 2.2.3 the following texts: "*Overall, this feature of separating herbaceous MTCs into multiple cohorts is coded more as a "place holder" for the current stage of model development rather than having solid scientific significance. To fully track soil carbon stocks of different vegetation types and their transient changes following land use change, a much larger number of cohorts are needed. But for a global application, this is limited by the computation efficiency.*"

line 285: "The cohort age subject to LUC of is one..." -> remove the of

done.

line 328: According to their webpage (http://gsweb1vh2.umd.edu/luh_data/LUHa.v1/readme.txt) LUH1 also makes a distinction of harvest from mature and young forest. Do youuse this information in your model, too? Furthermore, LUH contains "harvest from non-forested land", is this information used?

We treat harvest from mature forest in LUH1 as primary forest harvest, and this has already been explained in section 2.1.4 in the original manuscript (line 336). Harvest from young forest in LUH1 is implemented as secondary forest harvest (also see details in the same paragraph. "Harvest from non-forest land" is not included in our analysis. We inserted the following sentence in the 2[nd] paragraph of Sect. 2.2.3 to clarify this: "*Wood harvest from primary and secondary forests in LUH1 is used, while wood harvest from non-forest is not.*"

line 341: "first go first for" -> first go for

done.

line 347: should this maybe be secondary?

We indeed mean "primary harvest" here.

line 359: "to ensure the their" -> to ensure that their?

We changed "the" to "their".

line 386: but it respires in the grid cell where it is harvested?

Yes. Spatial relocation of harvested crops is not considered in the model. This point is now explained in the revised manuscript.

line 403: I do not understand this sentence

We apologize for this confusion. This sentence is changed to "$F_{Pasture}$ *for carbon sources from pastures other than harvest*".

line 427: remove the "and"?

done.

line 430: replace "on" with "by"?

done.

line 445: held constant or held as constants

It should be "held constant", now changed.

line 447: a hypothetical scenario

done.

line 448: I do not understand the sentence "Forest harvest of the same intensity..."

We mean "forest harvest of the same annual areal fraction". This is revised.

lines 556-561: But why is the NPP in simulations with age dynamics smaller? Is the forest in these simulations not yet as productive than intermediate-age forest?

For this particular case NPP is smaller with age dynamics, but the global run shows NPP with age dynamics is higher, in principle due to lower autotrophic respiration (because of slightly lower biomass) in the simulation with age dynamics. In general, as ORCHIDEE-MICT uses a big-leaf approximation that allows LAI to quickly level out and NPP reaching its maximum, and because cohort woody mass boundaries are the only parameter that differ among forest cohorts, we expect such differences in NPP between $S_{age}$ and $S_{ageless}$ to be subtle and do not have significant scientific implications in the ecological process. The difference in simulated $E_{LUC}$ is dominated by the difference in the forest biomass density being cleared. Therefore, the small differences in NPP between $S_{age}$ and $S_{ageless}$ have not been explored in depth.

line 702: Do you mean Hurtt et al. 2006? Else the reference is missing.

We mean Hurtt et al., 2006 and we apologize for this typo.

lines 710-715: It might be helpful to mention here again that LUH does include biomass harvest but that this is not used in your model.

This has been added.

line 748 this is section 6

This has been corrected.

line 753 and this section 7

This has been corrected.

line 1015: Fig. 9 does not include a "panel b"

This has been corrected.

---

## Author Response (AR2)

The topical editor finally decides the current Fig. 7 and its legend and caption are fine and no more revisions are needed.